# TumorChain: Interleaved Multimodal Chain-of-Thought Reasoning for Traceable Clinical Tumor Analysis

**Sijing Li**[1,2*], **Zhongwei Qiu**[2,3,1*], **Jiang Liu**[1], **Wenqiao Zhang**[1†], **Tianwei Lin**[1,2],
**Yihan Xie**[1], **Jianxiang An**[1], **Boxiang Yun**[2], **Chenglin Yang**[1], **Jun Xiao**[1],
**Guangyu Guo**[2,1], **Jiawen Yao**[2], **Wei Liu**[2], **Yuan gao**[2], **Ke Yan**[2], **Weiwei Cao**[2],
**Zhilin Zheng**[2], **Tony C. W. MOK**[2], **Kai Cao**[4], **Yu Shi**[5], **Jiuyu Zhang**[5], **Jian Zhou**[6],
**Beng Chin Ooi**[1], **Yingda Xia**[2†], **Ling Zhang**[2]
[1]Zhejiang University,  [2]DAMO Academy, Alibaba Group,  [3]Hupan Lab,
[4]Shanghai Institution of Pancreatic Disease  [5]Shengjing Hospital of China Medical University,
[6]Sun Yat-sen University Cancer Center
{suzylee, wenqiaozhang}@zju.edu.cn,
{qiuzhongwei.qzw, yingda.xia}@alibaba-inc.com

## Abstract

Accurate tumor analysis is central to clinical radiology and precision oncology, where early detection, reliable lesion characterization, and pathology-level risk assessment directly guide diagnosis, staging, and treatment planning. Chain-of-Thought (CoT) reasoning is particularly critical in this setting, as it enables step-wise interpretation from imaging findings to clinical impressions and pathology-level conclusions, ensuring traceability and reducing diagnostic errors. Here, we target the clinical tumor analysis task and build a large-scale benchmark that operationalizes a multimodal reasoning pipeline, spanning findings, impressions, and pathology predictions. We curate `TumorCoT`, a large-scale dataset of **1.5M CoT-labeled VQA instructions** paired with 3D CT scans, with step-aligned rationales and cross-modal alignments along the "findings → impression → pathology" trajectory, enabling standardized evaluation of both final accuracy and reasoning consistency. We further propose **TumorChain**, a multimodal interleaved reasoning framework that tightly couples 3D imaging encoders, clinical text understanding, and organ-level vision-language alignment. Through cross-modal alignment and iterative interleaved causal reasoning, TumorChain grounds visual evidence, aggregates conclusions, and issues pathology predictions after multiple rounds of self-refinement, improving traceability and reducing hallucination risk. TumorChain demonstrates consistent gains over strong unimodal and pipeline baselines in lesion detection, impression quality, and pathology classification, and successfully generalizes to the public DeepTumorVQA benchmark. Ablations validate the key contributions of interleaved reasoning and clinical CoT. Clinically, these advances lay the groundwork for reliable, interpretable tumor assessment to support real-world decision-making. To advance safe, explainable, and reproducible multimodal reasoning for high-stakes tumor analysis, detailed information about our project can be found on our project homepage at https://github.com/ZJU4HealthCare/TumorChain.

## 1 Introduction

Although large vision-language models (LVLMs) have achieved considerable success in general healthcare domains (Yang et al., 2024; Lin et al., 2025; Xu et al., 2025b; Lai et al., 2025), their application to multi-modal clinical understanding and reasoning remains challenging, especially in

---

*Equal contributions. The work was done during Sijing's internship at DAMO Academy.
†Corresponding author.

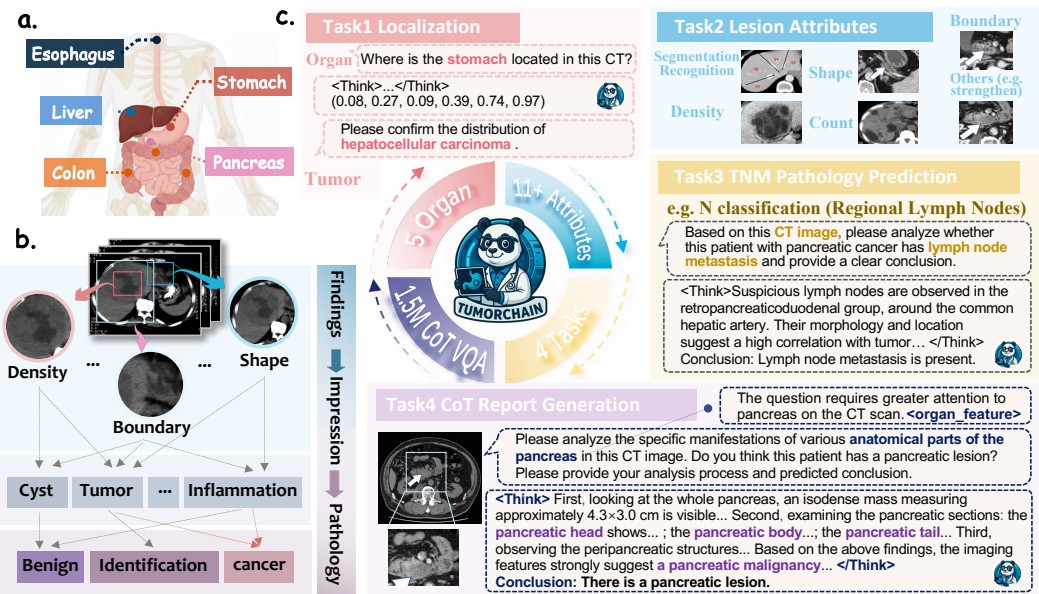

Figure 1: (a) **TumorChain** focuses on tumor analysis of the five major organs in the digestive system. (b) We establish a corresponding medical reasoning logic chain from radiology findings to radiology impressions to pathological conclusions. (c) The system tackles four diverse tasks across 1.5 million multimodal VQA instructions, enabling advanced clinical reasoning.

cancer discovery and risk assessment (Qiu et al., 2025) based on medical images. For example, oncologists need to identify suspected tumors on radiological images, distinguish benign from malignant lesions, and assess cancer risk and TNM [1] staging (Amin et al., 2017) combined with pathology report and other multidimensional patient features to perform treatment planning. Recent medical LVLMs (Med-LVLMs) primarily focus on study-level report generation. Nevertheless, these models exhibit notable limitations when applied to clinical oncologic scenarios: **(i) Limited Tumor-centric Specialization.** Current Med-LVLMs are optimized for broad clinical tasks (report generation, classification, coarse detection) (Bai et al., 2024; Xin et al., 2025; Chen et al., 2025b), but fall short in oncology-specific workflows: they do not reliably connect radiologic findings to pathology-level endpoints (TNM stage, nodal metastasis, risk stratification), and thus remain insufficient in clinical oncologic decision-making. **(ii) Scarcity of Diverse and Tumor-specific Datasets.** Tumor-related data are scarce and difficult to collect in existing medical datasets such as CT-RATE (Hamamci et al., 2024b) and 3D-RAD (Gai et al., 2025), which are designed as knowledge-constrained multiple-choice and short-text QA tasks and rarely support multi-granular analysis of individual cases. To this end, fine-grained visual representations of organ substructures and lesions are often misaligned with textual reasoning (Pan et al., 2025), increasing the risk of erroneous or fabricated medical conclusions. **(iii) Insufficient Reasoning Depth.** Most Med-LVLMs (Qiu et al., 2022; Wu et al., 2025; Xu et al., 2025b) are restricted to processing 2D medical images and rely on single-step reasoning for downstream tasks. In 3D radiologic scenarios, the greater structural and informational complexity makes single-step reasoning inadequate for multi-stage clinical inference. Additionally, the construction and assessment of reasoning chains remain insufficient.

To address these challenges, we build a large-scale, multi-institutional dataset targeting five major digestive organs (Fig. 1a)—and present a tumor-centric multimodal Chain-of-Thought (CoT) framework that operationalizes the full clinical pipeline from radiology findings and impressions to pathology-level diagnosis (Fig. 1b).

**(i) Dataset.** We collect multi-institutional 3D CT images from patients with pathology-confirmed tumors, each paired with the corresponding clinical radiology and pathology report . We also designed a multi-agent, knowledge-graph–guided engine that converts original reports into four CoT VQA tasks—localization, lesion attributes, TNM pathology prediction, and CoT report generation—yielding the largest known multimodal tumor-related dataset TumorCoT-1.5M with approximately 1.5M CoT VQA instruction samples(Fig. 1c).

---

[1]A staging system to describe the amount and spread of cancer in a patient's body, using TNM from AJCC.

**(ii) Benchmark.** We introduce a standard evaluation protocol that extracts "subject–relation–object" triplets (e.g. "pancreatic tail-discovered-malignant tumor") from CoT chains and performs stepwise scoring with an LLM-based evaluator. It measures the accuracy of abnormality detection and the logical correctness of downstream conclusions—from findings to impressions to pathology—providing a comprehensive assessment of clinical relevance and overall performance of Med-LVLMs.

**(iii) Model.** Technically, we propose **TumorChain**, a topology-aware, hybrid-model interleaved reasoning framework that integrates clinical priors and organ topology. As tumors frequently affect sub-structures and surrounding organs or distantly metastasize, TumorChain performs multi-round interleaved reasoning: the LLM ingests global tokens and the task prompt to surface intermediate "thoughts" and candidate organs; a segmentation expert returns masks and ROI-level tokens for the indicated regions; the augmented tokens are fed back to the LLM and the loop continues until no new ROIs emerge, producing a progressively refined CoT chain that fuses global context, local evidence, and task intent. To better optimize this framework, we adopt a collaborative hybrid-model joint training strategy: a segmentation expert produces spatial organ masks, a lightweight abnormality classifier built on local organ features enhances the visual encoder's organ-level abnormality discrimination, and the LLM performs multimodal integration and high-level clinical reasoning. This design enables topology-aware global–local fusion, reduces hallucinations, and improves end-to-end tumor assessment from findings to impressions to pathology.

The main contributions of this paper can be summarized as follows:

• **Clinical Tumor Reasoning Formulation**: We formulate the clinical tumor analysis task into a complete reasoning pipeline from radiology findings → study-level impressions → pathology predictions. This pipeline not only covers the core oncologic workflow, but also ensures traceability and interpretability, thereby reducing diagnostic errors.

• **Tumor CoT Data Engine and Evaluation**: We design an interactive-validated data engine and construct a large CoT-annotated VQA corpus (1.5M instances) for tumor understanding across five digestive organs, together with a CoT evaluation protocol that measures stepwise clinical reasoning.

• **Interleaved Multimodal Reasoning**: A medical multimodal interleaved-reasoning framework (**TumorChain**) with hybrid-model collaborative optimization (i.e., a segmentator, an abnormality classifier and an LLM) that achieves organ-level, global–local multimodal alignment for fine-grained 3D tumor analysis.

• **Substantial Performances and Insights**: Extensive experiments across downstream tumor tasks show consistent gains on the `TumorCoT-1.5M` test set and strong generalization to the public DeepTumorVQA (Chen et al., 2025a) benchmark. Our work provides actionable insights to develop multimodal foundation models in high-stakes clinical tumor analysis.

## 2 TUMORCOT-1.5M DATASET

### 2.1 DATA COLLATION AND ORGANIZATION

Most public medical datasets focus on general healthcare scenarios, featuring simple multiple-choice or short-answer formats lacking stepwise reasoning chains or region-specific causal annotations. To advance reliable tumor analysis with LVLMs, we curate `TumorCoT-1.5M`, a large-scale dataset centered on comprehensive tumor analysis in 3D CT, encompassing the entire clinical workflow from radiological findings and impressions to pathological prediction.

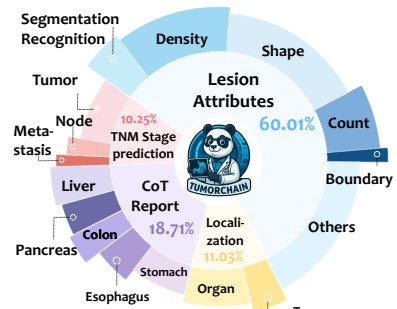

Figure 2: `TumorCoT-1.5M` statistics.

For data collection, we collaborated with multiple medical institutions to collect 3D CT scans from patients with tumors originated from five major digestive organs (liver, pancreas, stomach, colon, and esophagus). `TumorCoT-1.5M` includes approximately 41,059 CT images with corresponding captions, 10,708 radiology reports, and partial pathology reports. During the data partitioning stage, we strictly followed a patient-level protocol to divide the dataset into training and test sets at a ratio of 9:1, ensuring that there is no overlap between the two sets and preventing any data leakage.

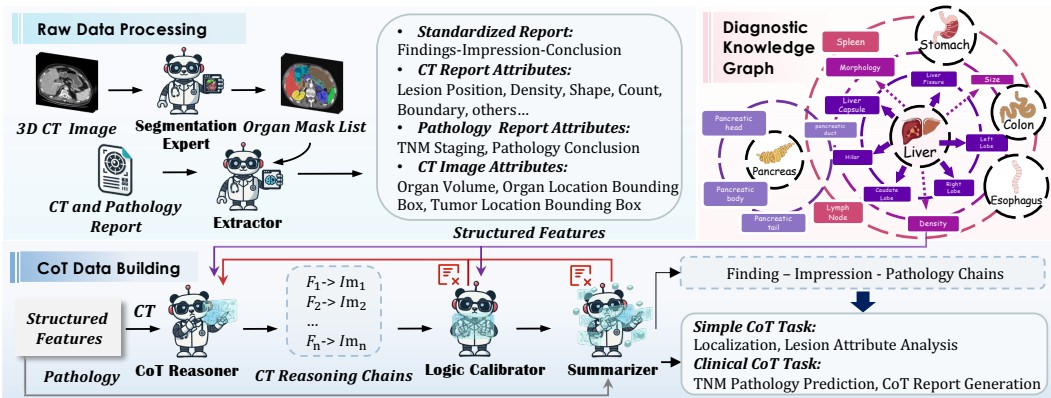

Figure 3: The overview of the interactive-validated CoT Data Engine, which is a system of multi-agent collaboration, including the procedure of raw data processing, diagnostic knowledge graph importing, and CoT data building and validation.

To fully leverage tumor CT data from diverse sources and formats, we developed an interactive-validated CoT data engine to construct `TumorCoT-1.5M`. In addition, panels of radiology and pathology experts conducted rigorous reviews of the reasoning chains(details see Appendix E.5), supplementing missing links, and ensuring logical consistency and traceability at every step.

Ultimately, as shown in Figure 2, `TumorCoT-1.5M` comprises 1,497,818 CoT-VQA pairs, covering four key tasks in tumor analysis: **(i) Localization**, **(ii) Lesion Attribute Analysis**, **(iii) TNM Stage Prediction**, and **(iv) CoT Report Generation**. Each type is further divided into subtasks based on organ substructures or tumor grades, with tasks ranging from simple to complex in both multiple-choice and open-ended formats. Every sample includes a traceable reasoning process and summary. This organizational framework markedly improves the model's reasoning interpretability and cross-modal consistency.

## 2.2 COT DATA ENGINE

We designed an interleaved-validation CoT data engine to construct `TumorCoT-1.5M`. Guided by an expert-level knowledge graph, we generated diverse fine-grained CoT-VQA samples across three major steps in tumor analysis: ROI localization, CT interpretation, and pathology prediction. As shown in Figure 3, the data engine comprises the following 3 procedures with 6 agentic experts (detailed in Appendix E and details of the 6 experts are in Appendix E.1):

**(i) Raw Data Processing**: This process includes two expert agents: the Segmentation Expert Model (TotalSegmentator Wasserthal et al. (2023)) and the Structured Feature Extractor (Qwen3-235B-A22B (Yang et al., 2025)), aiming to clean data, standardize terminology, obtain organ masks(Appendix E.4), and extract structured features from CT and pathology reports.

**(ii) Diagnostic Knowledge Graph-Driven Prompt Engineering**: In collaboration with radiologists and pathologists, we construct diagnostic knowledge graphs (KGs) covering five major organs (The KG for 5 digestive organs is shown in Appendix E.2). During reasoning chain construction, relevant nodes and relationships are retrieved and then authoritative clinical guidelines are integrated into the LLM prompts to structurally constrain the model to follow professional medical standards, thereby ensuring traceability and high reliability of the logical chain.

**(iii) CoT Data Building and Cross-Review Mechanism**: This process includes 3 agents: **CoT Reasoner**, **Logic Calibrator**, and **Summarizer**. To fully utilize the strengths of different LLMs, three agents are assigned distinct models. Additionally, we use only the textual content from the reports as their sole source of information, thereby preventing these models from making subjective assumptions about complex modalities such as CT imaging. CoT Reasoner uses GPT-4o-mini for its strong language and medical capabilities, processing structured CT report features and integrating external KGs to link radiological findings with impressions. Logic Calibrator, based on Claude3.5-Haiku, automatically validates reasoning accuracy. If potential issues are detected, the system randomly applies two prompting strategies—expanding the organ region or providing a sus-

pected cause—to guide the reasoning model in re-evaluating its chain. Summarizer employs GPT-5-mini to process structured pathology reports, aggregate and verify reasoning chains, and generate QA pairs and CoT-formatted reports when chains align with pathological conclusions; otherwise, it triggers upstream re-reasoning. The agents' prompts are shown in Appendix E.3.

## 2.3 CLINICAL CoT EVALUATION: TUMORCHAIN-EVAL

Existing evaluation metrics like accuracy, semantic similarity metric (Rouge-L (Lin, 2004), CIDEr Vedantam et al. (2015), RaTEScore Zhao et al. (2024), etc.) ignore the correctness of clinical logic and step-wise evaluation of the CoT reasoning process. To address this issue, we propose **TumorChain-Eval**, a novel evaluation framework with metric $CoT_e$ specifically for oncology-related CoT reasoning, which provides a fine-grained and interpretable scoring system based on reasoning logic chains. In this approach, the CoT reasoning process of the ground truth and prediction is extracted into three reasoning chains, all of which can be represented as subject-relation-object triplets: **(i) Finding Chain**: Describes the primary radiology findings, which are independent facts and do not involve higher-order reasoning. **(ii) Impression Chain**: Summarizes intermediate-level clinical impressions based on the Finding Chain, involving simple reasoning conclusions from multiple findings. **(iii) Long Reasoning Chain**: Combines multiple findings and impressions to generate higher-order reasoning, such as conclusive diagnostic hypotheses with strong logical inference. Each type of chain is evaluated using a detailed set of scoring criteria, and the scores are calculated by using GPT-4 with scoring guidelines. Finally, the $CoT_e$ score is computed by weighted sum of $S_{FC}$, $S_{IC}$, $S_{LRC}$, which are the FC score, LC score, and LRC score, respectively. The detailed scoring process and criteria for each reasoning chain can be found in the Appendix F.

## 3 METHODOLOGY

### 3.1 OVERVIEW OF TUMORCHAIN

Before presenting our method, we first introduce some basic notions and terminologies. The framework $\mathcal{F}(\cdot)$ of TumorChain is shown in Figure 4, which consists of five key modules: a 3D vision encoder $\mathcal{E}_v(\cdot)$, an organ segmentation expert $\mathcal{S}eg(\cdot)$, an auxiliary classification model $\mathcal{C}ls(\cdot)$, a multi-layer perceptron (MLP) projector $\mathcal{P}(\cdot)$, and a LLM $\mathcal{LLM}(\cdot)$. Given the CT volumes $\mathcal{V}_{ct} \in \mathbb{R}^{H \times W \times D}$ with task prompt text $\mathcal{T}_{task}$ as input, the TumorChain outputs the response $\mathcal{R}_{cot}$ of the interleaved multi-modal chain of thought (CoT), which simplifies as below:

$$\underbrace{\mathcal{F}(\mathcal{V}_{ct}, \mathcal{T}_{task})}_{\textbf{TumorChain}} : \underbrace{\mathcal{E}_v(\cdot) \to \mathcal{S}eg(\cdot) \to \mathcal{C}ls(\cdot) \to \mathcal{P}(\cdot)}_{\textbf{Global\&Local Visual Alignment}} \to \underbrace{\mathcal{LLM}(\cdot) \to \mathcal{R}_{cot}}_{\textbf{Interleaved Multi-modal Inference}} \quad (1)$$

**Global & Local Visual Alignment.** For global visual tokens, the 3D encoder $\mathcal{E}_v$ takes the $V_{ct} \in \mathbb{R}^{H \times W \times D}$ as input that encode 3D CT volumes to a series of vision tokens as $\tau_v = \mathcal{E}_v(V_{ct})$. Then, all vision tokens are aligned into LLM space by the projector $\tau_g = \mathcal{P}(\tau_v)$, where $\tau_g$ denotes the global tokens and has a size of $L_g \times K$ ($L_g$ tokens and embedding dimension K).

For local visual tokens, the organ segmentation expert $\mathcal{S}eg(\cdot)$ first segments the organ mask as $\mathcal{M}_{organ} = \mathcal{S}eg(V) \in \mathbb{R}^{H \times W \times D}$ to provide fine-grained location for interlaced reasoning. According to the task prompt, TumorChain identifies the target task organ mask $\mathcal{M}_{task}$ from $\mathcal{M}_{organ}$ by matching the organ in the task text and extracts local vision tokens $\tau_l$ as

$$\tau_l = \Gamma(\tau_g, \mathcal{M}_{task}) \quad s.t. \quad \mathcal{M}_{task} = \Lambda(\mathcal{M}_{organ}, \mathcal{T}_{task}), \quad (2)$$

where $\Lambda(\cdot)$ denotes the organ matching operation and $\Gamma(\cdot)$ represents the operation of extracting local token $\tau_l$ of size $L_l \times K$ according to mask.

Besides, to enhance the $\tau_l$ learning, we introduce the auxiliary classification model $\mathcal{C}ls(\cdot)$ that projecting $\tau_l$ into two classification logits, which can be formulated as $y = Cls(\tau_l)$, where the $\mathcal{C}ls(\cdot)$ distinguish normal or abnormal for each local organ, thereby significantly enhancing the encoder's discriminative capability. When the visual features and task prompts are jointly input into the LLM, the model's attention to key information and anomalous regions can be further enhanced.

**Interleaved Multi-modal Inference.** After obtaining all vision and text tokens, they are combined into interleaved tokens to serve as the input for LLM, to further generate the inference response $\mathcal{R}_{cot}$

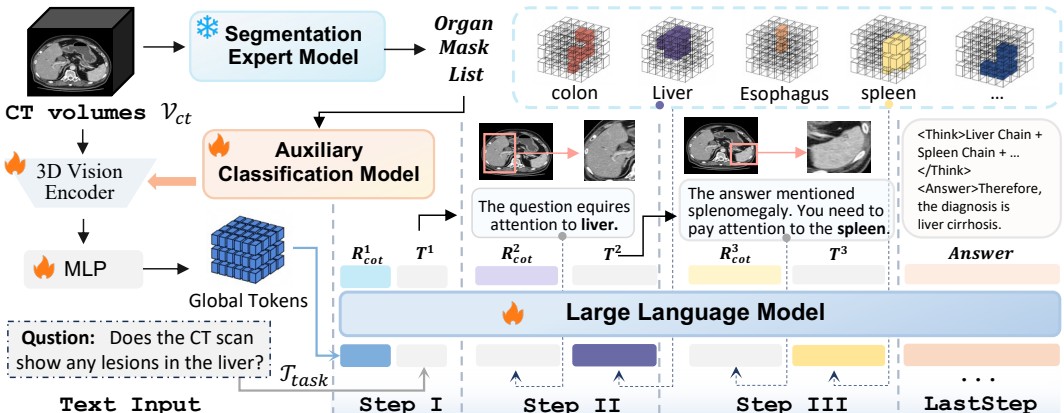

Figure 4: The overview of **TumorChain**. Given CT volumes and task prompts, TumorChain analyzes the task requirements and performs iterative interleaved reasoning by integrating a segmentation expert model for ROI localization, auxiliary classification for local feature enhancement, and organ-guided iterative reasoning for clinical tumor analysis.

of CoT as

$$\mathcal{R}_{cot} = \mathcal{LLM}(\tau_{in}) \quad s.t. \quad \tau_{in} = [\tau_g, \mathcal{T}_{task}, \mathcal{T}^1, \tau_l^1, \mathcal{T}^i, \tau_l^i, ...], \tag{3}$$

where $\tau_{in}$ indicates all input tokens of LLM, and is the combination of global vision tokens $\tau_g$, task text prompt tokens $\mathcal{T}_{task}$, and task target organ tokens $\tau_l^i$ with its text pair $\mathcal{T}^i$. Here $i \in \{1, \ldots, \mathcal{O}\}$ indexes the $\mathcal{O}$ potential related or ROI organs/suborgans reasoned by LLM.

Through the aforementioned process, TumorChain achieves interleaved alignment of global visual features, task text, and task-specific local visual features during a single inference by leveraging the collaborative interaction between large and small models.

## 3.2 ORGAN-GUIDED ITERATIVE INTERLEAVED REASONING

Clinical 3D tumor analysis demands organ identification, fine-grained tumor localization, and reliable ROI evidence for LLM reasoning, making single-pass inference insufficient and complicating multimodal self-validation. To achieve a reliable clinical reasoning chain of "finding → impression → pathology prediction", TumorChain adopts an **organ-guided Iterative Interleaved Reasoning (IIR)**. IIR uses each LLM output to refine ROI selection, feeds the resulting ROI features back for the next round, and repeats, which enables multi-round interleaved reasoning with built-in self-validation that maintains visual–text alignment across iterations. For clarity, we abstract the IIR process into three steps:

**Step I: Initial Round of Reasoning.** The initial reasoning step takes the encoded global CT volume tokens $\tau_g$, and the medical question text $T_{task}$, as the inputs to the LLM:

$$\mathcal{R}_{cot}^1 = \mathcal{LLM}(\tau_g, T_{task}), \tag{4}$$

where $\mathcal{R}_{cot}^1$ represents the initial diagnostic result output by the LLM.

**Step II: Interleaved Self-reflection and Organ Localization.** *(i) Keyword Extraction and Organ Identification:* According to the initial output $R_{cot}^1$ and Equation 2, determining the local ROI mask $M_l^1$ to obtain local vision tokens $\tau_l^1$. *(ii) Augmented Prompt Construction:* After generating organ-specific tokens based on the extracted organ features, we then construct augmented prompts as:

$$\mathcal{T}_l^1 = (\text{"The question requires greater attention to"}, organ_l^1), \tag{5}$$

where $organ_l^1$ represents the identified ROI from $R_{cot}^1$. *(iii) Combining Inputs for the Next Iteration:* Then, the original question $\mathcal{T}_{task}$, the initial answer $R_{cot}^1$, and identified organ tokens $\tau_l^1$ are combined to form a new input for the LLM:

$$\tau_{in}^2 = [\tau_g, \mathcal{T}_{task}, \mathcal{R}_{cot}^1, \mathcal{T}_l^1, \tau_l^1]. \tag{6}$$

**Step III: Iterative Causal Reasoning**   After completing the previous self-reflection, the framework continues to perform iterative reasoning based on the newly generated answers. If additional relevant organs are introduced, the ROI extraction process is repeated to sequentially obtain the features of all organs involved in the reasoning chain, supporting further verification. According to Equation 3 and 6, this iterative process can be formulated as

$$\mathcal{R}_{cot}^{i+1} = \mathcal{LLM}(\tau_g, \mathcal{T}_{task}, \mathcal{R}_{cot}^i, \mathcal{T}^i, \tau_l^i). \tag{7}$$

For example, in cases where cirrhosis of the liver is suspected and the imaging findings indicate splenomegaly and altered hepatic lobe proportions, IIR automatically extracts the organ tokens for the spleen and appends a pertinent prompt before it feeds both the original question and the previous answers into the LLM. The framework undertakes multi-round causal verification of each organ in the answer until all relevant organs have undergone feature extraction and causal reflection.

## 3.3 HYBRID-MODEL COLLABORATIVE OPTIMIZATION STRATEGY

Typically, a 3D CT scan contains over 100 slices, each with a resolution of $512 \times 512$ pixels. In comparison to 2D image-caption pairs (e.g., resolutions of 224 or 384), this makes it challenging to achieve fine-grained alignment between the visual representations and textual descriptions. To address the above challenges and further optimize TumorChain, we employ a Hybrid-model Collaborative Optimization (HCO) strategy. The core concept of HCO lies in the collaboration of models at different scales: the segmentation model localizes ROIs, continuously providing precise local features for reasoning. The classification model ensures discriminative local feature learning during optimization, enabling the visual encoder to effectively distinguish abnormal from normal patterns, preventing subtle anomalies from being overshadowed during LLM training. The LLM integrates reasoning outcomes and leverages the segmentation model for iterative decision-making.

During training, the classification model and LLM are jointly optimized via loss functions. The training loss $L_{total}$ is:

$$L_{total} = -\frac{1}{N} \sum_{n=0}^{N} log P(r_n | \tau_{in}, r_1, ..., r_{n-1}) - \alpha \cdot \frac{1}{M} \sum_{m=0}^{M} \hat{y} P(y | \tau_l), \tag{8}$$

where $N$ represents the output text length and $M$ represents the sample number. $r_i \in R^N$. $\alpha$ is a loss weight. $y$ and $\hat{y}$ are the prediction and ground-truth of classification (normal/abnormal) for local organs. During inference, the segmentation model utilizes the ROI organ locations provided by the LLM to extract ROI visual features, which are then fed back into the LLM for self-verification.

## 4 EXPERIMENTS

### 4.1 EXPERIMENTAL SETTINGS

**Model Overview.**   Both TumorChain-3B and TumorChain-7B employ the pretrained LLM backbone of Qwen2.5-VL-3B/7B as the large language model component, integrating M3D as the vision encoder (Bai et al., 2024) to enable 3D CT feature extraction. Full architectural details and training configuration are provided in Appendix D.1.

**Baselines.**   To comprehensively evaluate the performance of TumorChain, we compare it against a diverse set of baseline models, including seven open-world LVLMs (e.g., Claude 3 (Anthropic, 2024), Gemini 2.0 (DeepMind, 2025), GPT-5 (OpenAI, 2025), Qwen2.5-VL (Bai et al., 2025), InternVL-2.5 (Chen et al., 2024), MiniCPM-V4.5 (Yu et al., 2025b), LLaVA-CoT (Xu et al., 2025a)), three 2D-based Med-LVLMs (e.g., HealthGPT (Lin et al., 2025), Lingshu (Xu et al., 2025b), MedVLM-R1 (Pan et al., 2025)), and two 3D-based Med-LVLMs ( e.g., RadFM (Wu et al., 2025), M3D-Phi-3 (Bai et al., 2024)).

**Benchmarks.**   All models are evaluated on the test subset of TumorCoT-1.5M, partitioned via a 9:1 split. Evaluation is conducted based on our definitions of four main tasks and twelve subtask types.

**Evaluation Metrics.**   We developed diverse protocols focusing on the analytical process and conclusions within the reasoning chain. All models are guided to generate CoT reasoning chain, which

Table 1: Comparison of TumorChain with other LVLMs on `TumorCoT` benchmark. **Bold** and underlined text indicates the best and second-best performance, respectively.

| Model | Position | | Lesion Attributes | | | | | | TNM Prediction | | | CoT-Report | Avg. |
|---|---|---|---|---|---|---|---|---|---|---|---|---|---|
| | Organ Pos. | Tumor Pos. | Seg. Loc. | Shape | Boundary | Density | Count | Others | Tumor | Node | Met. | | |
| *Commercial Models* | | | | | | | | | | | | | |
| Claude3-Haiku | 32.61 | 33.05 | 52.38 | 58.51 | 50.00 | 50.74 | 37.51 | 50.95 | 59.19 | 56.59 | 43.40 | 33.21 | 46.51 |
| Gemini2.0-Flash | 43.27 | 17.69 | 46.81 | 35.97 | 38.22 | 39.11 | 40.22 | 52.86 | 34.03 | 42.21 | 49.06 | 56.08 | 41.29 |
| GPT-5-Mini | 44.18 | 37.73 | 44.96 | 47.72 | 62.58 | 57.13 | 64.85 | 66.67 | 34.25 | 47.40 | 49.60 | 62.02 | 51.59 |
| *Generalist Models* | | | | | | | | | | | | | |
| Qwen2.5-VL-7B | 29.57 | 32.52 | 27.41 | 43.96 | 57.51 | 49.36 | 46.73 | 58.02 | 34.17 | 37.49 | 51.15 | 60.54 | 44.04 |
| InternVL-2.5-8B | 29.10 | 27.67 | 33.50 | 49.96 | 51.45 | 49.48 | 42.68 | 51.70 | 36.16 | 35.07 | 48.09 | 52.18 | 42.25 |
| MiniCPM-V4.5-9B | 35.70 | 29.56 | 44.11 | 52.61 | 53.67 | 50.80 | 46.92 | 58.45 | 43.12 | 44.74 | 44.27 | 47.84 | 45.98 |
| LLaVA-CoT-11B | 31.63 | 27.64 | 37.70 | 59.44 | 49.34 | 49.60 | 37.90 | 55.48 | 48.20 | 47.91 | 49.23 | 44.81 | 44.91 |
| *2D Medical Models* | | | | | | | | | | | | | |
| HealthGPT-3.8B | 30.58 | 22.80 | 45.87 | 51.35 | 45.77 | 47.27 | 31.24 | 40.13 | 45.68 | 49.02 | 39.69 | 39.20 | 40.72 |
| Lingshu-7B | 25.28 | 32.58 | 24.90 | 51.70 | 60.61 | 53.40 | 45.21 | 63.63 | 30.35 | 42.79 | 56.49 | 54.30 | 45.10 |
| MedVLM-R1-2B | 32.23 | 38.92 | 19.37 | 48.79 | 51.20 | 48.02 | 44.53 | 52.26 | 30.38 | 27.26 | 32.82 | 39.70 | 38.79 |
| *3D Medical Models* | | | | | | | | | | | | | |
| RadFM | 13.86 | 15.97 | 15.23 | 21.25 | 49.84 | 21.07 | 19.34 | 24.15 | 49.58 | 46.02 | 28.47 | 21.34 | 26.32 |
| M3D-Phi-3-4B | 32.39 | 28.28 | 22.69 | 23.93 | 42.74 | 30.49 | 27.19 | 52.35 | 44.82 | 30.80 | 35.86 | 34.79 | 32.84 |
| *Our Models* | | | | | | | | | | | | | |
| **TumorChain-3B** | 99.93 | 96.44 | 79.13 | 69.07 | 73.57 | 68.97 | 81.04 | 76.34 | 86.34 | 55.09 | 56.49 | 78.96 | 76.78 |
| **TumorChain-7B** | **99.97** | **97.57** | **86.88** | **82.28** | **84.52** | **85.05** | **86.20** | **86.57** | **88.83** | **61.63** | **71.07** | **82.36** | **84.41** |
| *w/o CoT, IIR* | 99.92 | 97.45 | 78.95 | 75.78 | 74.60 | 80.18 | 82.82 | 78.75 | 84.37 | 59.29 | 66.85 | 79.83 | 79.90 |
| *w/o CoT* | 99.91 | 97.10 | 82.75 | 78.43 | 82.57 | 82.79 | 84.26 | 84.16 | 88.34 | 60.76 | 68.12 | 80.26 | 82.45 |
| *w/o IIR* | 99.87 | 97.55 | 79.36 | 79.52 | 74.62 | 81.46 | 80.60 | 79.20 | 84.65 | 60.32 | 65.87 | 81.06 | 80.34 |

is scored using our proposed TumorChain-Eval. For conclusion, accuracy is calculated for multiple-choice tasks (e.g., segment localization in Task 2 and Task 1). For open-ended QA tasks, semantic consistency of the conclusions is assessed using GPT-5 and semantic accuracy is computed. Our evaluation prompt design and implementation details are provided in Appendix D.

## 4.2 MAIN RESULTS

**CoT-Conclusion Analysis.** We evaluated both the reasoning process and the correctness of conclusions for our four proposed tumor analysis tasks. The experimental results for conclusion accuracy are presented in Table 1, lines 1 to 13, and organ-level accuracy is displayed in Figure 5b. The main observations are as follows: **(i) Superior Performance.** TumorChain-7B achieves state-of-the-art results across all subtasks, with an average accuracy of 84.41%, substantially outperforming all baseline models and. TumorChain-3B also demonstrates strong performance, highlighting the effectiveness of our dataset and model design in complex tumor analysis scenarios. **(ii) Limitations of Existing Med-LVLMs.** Current mainstream 2D or 3D medical models do not surpass, and in some metrics even underperform, compared to general-purpose or commercial models in tumor analysis. This indicates the limited generalization and adaptability of existing Med-LVLMs, especially in complex, disease-specific settings, and in unseen datasets. Notably, some 3D medical models perform significantly worse than 2D models on CT analysis, revealing critical bottlenecks in spatial understanding and multi-attribute analysis of large-scale CT scans. **(iii) Effectiveness of Multi-granularity Benchmark Design.** The performance of our models across various subtasks demonstrates the multi-level and fine-grained nature of our dataset.

Near-perfect accuracy in basic localization tasks and relatively lower scores in more challenging pathological prediction and report generation emphasize the comprehensive and progressive challenges posed by our benchmark, providing valuable insights for fine-grained AI development in the domain of tumor analysis and diagnosis.

**CoT-Reasoning Analysis.** We compare CoT reasoning ability with other reasoning models in Table 2 via the proposed $CoT_e$ metric (Section 2.3). TumorChain-7B demonstrates high accuracy in generating traceable reasoning (FC, IC) and complex reasoning chains (LRC), achieving $CoT_e$ scores of 58.33 (see Figure 5c for example). Its clinical reasoning ability outperforms

Table 2: TumorChain-Eval with other **reasoning LVLMs** on `TumorCoT`.

| Reasoning Models | $S_{FC}$ | $S_{IC}$ | $S_{LRC}$ | $CoT_e$ |
|---|---|---|---|---|
| *Commercial Models* | | | | |
| Claude3-Haiku | 59.31 | 43.66 | 30.80 | 43.21 |
| Gemini2.0-Flash | 63.10 | 57.69 | 45.12 | 54.28 |
| GPT-5-Mini | **64.22** | **66.42** | **55.09** | **61.23** |
| *Open-Source Models* | | | | |
| Qwen2.5-VL-7B | 49.15 | 22.89 | 12.09 | 26.45 |
| InternVL-2.5-8B | 56.66 | 57.92 | 47.05 | 53.19 |
| MiniCPM-V4.5-9B | 60.51 | 58.52 | 45.44 | 53.89 |
| HealthGPT-3.8B | 49.64 | 49.68 | 37.15 | 44.65 |
| Lingshu-7B | 52.51 | 31.93 | 16.46 | 31.92 |
| MedVLM-R1-2B | 40.25 | 36.05 | 21.23 | 31.38 |
| **TumorChain-3B** | 61.75 | 49.99 | 35.68 | 47.98 |
| **TumorChain-7B** | **62.37** | **62.60** | **52.57** | **58.33** |

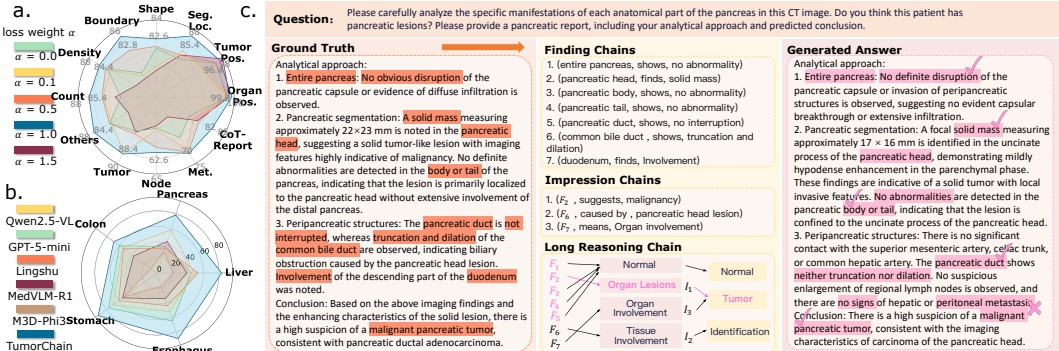

Figure 5: (a) Ablation on classification loss weight $\alpha$. (b) Model performance on five organs of TumorCoT. (c) Workflow of reasoning chains extraction and evaluation for task 4.

the general LVLMs (e.g., Qwen2.5-VL-7B) and Med-LVLMs (e.g., Lingshu-7B), which are trained on more medical data with diverse modalities. Furthermore, TumorChain shows better pxunrformance than commercial models (Claude3-Haiku, Gemini2.0-Flash). Nevertheless, the $CoT_e$ score is slightly lower than the performance of the latest GPT5-mini, demonstrating its strong capabilities in the medical domain.

## 4.3 ABLATION AND GENERALIZATION STUDIES

**Effect of CoT and IIR.** To comprehensively evaluate the impact of our proposed CoT reasoning and Interleaved Iterative Reasoning (IIR) approaches, we conducted systematic ablation studies. As shown in Table 1, lines 13 to 16, the combined use of CoT and IIR yields substantial performance gains across all subtasks, with an overall accuracy improvement of 5.64% compared to the baseline. This highlights that focusing on ROI region features markedly enhances model performance in 3D image analysis. Models employing CoT alone consistently improved results on most tasks, notably advancing clinical interpretability without sacrificing analytical accuracy. We further assessed inference latency after incorporating the IIR mechanism. The segmentation model incurs only a minimal increase in inference time (2.51 seconds per sample), while delivering an approximately 4% accuracy boost. The significant benefit of emphasizing ROI features far outweighs the slight time cost, strongly validating the reliability and effectiveness of our method for intelligent tumor analysis.

**Impact of Classification Loss Weight.** We systematically investigated the model's performance under different classification loss weight settings, as illustrated in Figure 5 (a). The results indicate that appropriately increasing the classification loss weight (e.g., $\alpha$=1.0) leads to significant performance improvements across multiple tasks, achieving the highest average accuracy of 84.41%. However, setting the weight too low or too high (such as $\alpha$=0.0 or $\alpha$=1.5) results in diminished performance, further demonstrating the essential role of loss weight adjustment in the collaborative optimization of hybrid models. Detailed results are available in Appendix D.3.

**Effect of finetuning by TumorCoT-1.5M train set.** To more directly demonstrate the independent contribution of our dataset to model finetuning, we selected three representative baseline models for finetuning on our TumorCoT-1.5M training set: the 3D medical LVLM M3D-Phi-3-4B, the 2D medical LVLM Lingshu-7B, and the 2D general-purpose model Qwen2.5-VL-7B. Figure 6 shows the performance changes of these baseline models before and after finetuning (see Appendix D.3.3 for detailed results). The results indicate that all baseline models achieve significant improvements after finetuning, demonstrat-

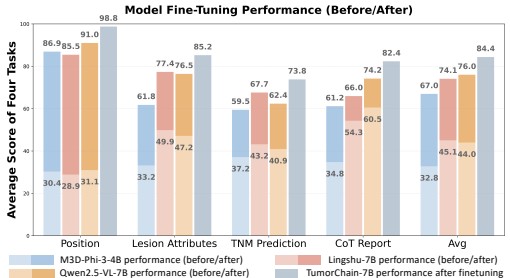

Figure 6: Baseline model improvements by finetuning with TumorCoT-1.5M.

ing the unique and important contribution of our dataset in the tumor domain. Meanwhile, our proposed TumorChain-7B still substantially surpasses the finetuned baselines, further highlighting the effectiveness of our model design.

**Generalization on DeepTumorVQA Benchmark.** To evaluate the generalization capability of our model and methods in tumor analysis, we conducted comprehensive experiments on the latest public tumor benchmark, DeepTumorVQA. As shown in Table 3, without seeing any data from this dataset, TumorChain-7B achieved an accuracy of 73.30% in lesion recognition and demonstrated substantially better performance in both visual and medical reasoning compared to leading commercial and open-source models. In addition, its average accuracy exceeds that of the second-best reasoning model, MedVLM-R1, by 14.84%. These results strongly demonstrate the exceptional generalization ability and robustness of our approach in open-domain tumor analysis tasks.

Table 3: Results on DeepTumorVQA.

| Model | Recog. | V.R. | M.R. | Avg. |
|---|---|---|---|---|
| *Commercial Models* | | | | |
| Claude3-Haiku | 38.70 | 31.53 | 34.81 | 35.01 |
| Gemini2.0-Flash | 50.88 | 33.81 | 35.06 | 39.92 |
| GPT-5-mini | 49.29 | 35.57 | 29.87 | 38.24 |
| *Open-Source Models* | | | | |
| Qwen2.5-VL-7B | 45.31 | 36.94 | 37.04 | 39.76 |
| InternVL-2.5-8B | 50.06 | 31.35 | 44.20 | 41.87 |
| MiniCPM-V4.5-9B | 50.14 | 33.69 | 42.86 | 42.23 |
| HealthGPT-3.8B | 43.32 | 30.68 | 39.75 | 37.92 |
| Lingshu-7B | 50.24 | 36.19 | 37.78 | 41.40 |
| MedVLM-R1-2B | 56.41 | 38.27 | 33.33 | 42.67 |
| RadFM | 50.19 | 39.82 | 30.87 | 40.29 |
| M3D-Phi-3-4B | 29.53 | 23.22 | 23.70 | 25.48 |
| **TumorChain-7B** | **73.30** | **53.31** | **45.93** | **57.51** |

## 5 RELATED WORK

**Medical Large Vision-Language Models.** Med-LVLMs such as LLaVA-Med (Li et al., 2023), Lingshu (Xu et al., 2025b) and HealthGPT (Lin et al., 2025) have rapidly advanced multimodal clinical decision support by aligning ViT-based 2D image features with language models. Methods like M3D (Bai et al., 2024), CT-Chat (Hamamci et al., 2024a) and Merlin (Blankemeier et al., 2024) incorporate 3D CNNs and ViTs to better capture volumetric data (Qiu et al., 2023), enabling more precise analysis of clinical 3D images. However, these Med-LVLMs lack causal reasoning for complex cases, prompting a shift toward CoT and knowledge-guided reasoning paradigms. Med-VLThinker (Huang et al., 2025) and MedVLM-R1 (Pan et al., 2025) combine CoT reasoning with reinforcement learning to improve interpretability, while ReasonMed (Sun et al., 2025) optimizes reasoning paths via multi-agent verification. MedResearcher-R1 (Yu et al., 2025a) enhances deep reasoning by leveraging knowledge graphs for multi-hop QA sample construction. By emphasizing logical chains, medical vision reasoning models are overcoming traditional 2D and 3D VLM limitations and opening new directions for explainable and reliable clinical decision-making. We further summarize related work with our TumorChain in three aspects: **Medical Benchmark**, **Medical Large Vision-Language Models**, and **Online Interleaved Reasoning**. The detailed discussions are shown in Appendix C.

## 6 CONCLUSION

We introduce **TumorChain**, a multimodal framework designed for comprehensive clinical tumor reasoning, spanning radiological findings, study-level impressions, and pathology predictions. We also construct `TumorCoT-1.5M`, the largest multimodal dataset for tumor reasoning, and propose a benchmark protocol to evaluate stepwise reasoning fidelity. Extensive experiments demonstrate TumorChain's superior performance across diverse tumor-related tasks, benefiting the advancement of clinical AI in oncology. Furthermore, Appendix G presents case studies and qualitative error analyses as useful references for future model improvement and clinical practice.

## ACKNOWLEDGMENTS

This work has been supported in part by the NSFC (No. 62436007), the China Postdoctoral Science Foundation under Grant Number 2024M752794, the ZJNSF (No. LZ25F020004), the Key Research and Development Projects in Zhejiang Province (No. 2025C01128, 2025C01030, 2025C02156), Ningbo Yongjiang Talent Introduction Programme (2023A400-G).

## ETHICS STATEMENT

This study used hospital-held clinical imaging data and was reviewed and approved by the hospital Ethics Committee. Data collection and use complied with applicable laws, regulations, and ethical standards. All data were de-identified and anonymized prior to use, with personally identifiable information removed. Access to the data was restricted to authorized personnel, and appropriate safeguards (e.g., encryption and access controls) were implemented to prevent disclosure of personal information. The data were used solely for the purposes of this research.

## REPRODUCIBILITY STATEMENT

To advance safe, explainable, and reproducible multimodal reasoning for high-stakes tumor analysis, detailed information about our project can be found on our project homepage at https://github.com/ZJU4HealthCare/TumorChain. Detailed descriptions of hyperparameters and experimental settings can be found in Appendix D.1.

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

APPENDIX

## A LLM Usage Statement

In this work, we employ LLMs in three strictly controlled ways. First, LLMs are utilized to rephrase the source data during dataset construction in the CoT data engine E), , thereby constructing a high-quality training dataset based on the original information while preserving the clinical context. Second, we leverage LLMs as experts to evaluate our model against baselines (see Appendix F), following clearly defined criteria for consistency and fairness. Third, we employ LLMs to identify and correct grammatical errors in our manuscript, ensuring clarity and linguistic accuracy throughout the paper.

## B Notation Table

To facilitate understanding and ensure consistency of symbols used throughout the paper of Tumor-Chain , Table 4 provides a concise summary of all key notations, offering readers a quick reference for the variables and operators involved.

Table 4: Notation in the TumorChain Pipeline.

| Notation | Description |
|---|---|
| $\mathcal{F}(\cdot)$ | Represents the TumorChain framework. |
| $\mathcal{E}_v(\cdot)$ | 3D vision encoder |
| $\mathcal{S}eg(\cdot)$ | Organ segmentation expert |
| $\mathcal{C}ls(\cdot)$ | Auxiliary classification model |
| $\mathcal{P}(\cdot)$ | Multi-layer perceptron (MLP) projector |
| $\mathcal{LLM}(\cdot)$ | Backbone of LLM |
| $\mathcal{V}_{ct} \in \mathbb{R}^{H \times W \times D}$ | Represents a 3D CT volume of height $H$, width $W$, and depth $D$. |
| $\mathcal{T}_{task}$ | Input of task prompt text |
| $\mathcal{R}_{cot}$ | The output of TumorChain |
| $\tau_v$ | A series of vision tokens after 3D encoder $\mathcal{E}_v$ |
| $\tau_g$ | Global CT volume tokens (projector-aligned into LLM space). |
| $\mathcal{M}_{organ}$ | Denotes the organ mask generated by the segmentation model $\mathcal{S}eg(\cdot)$ |
| $\tau_l$ | Local vision tokens |
| $\Lambda(\cdot)$ | Denotes the organ matching operation |
| $\Gamma(\cdot)$ | Represents the operation of extracting local token $\tau_l$ of size $L_l \times K$ according to mask. |
| $y$ | Result after Classification layer $Cls(\tau_l)$ |
| $\tau_{in} = [\tau_g, \mathcal{T}_{task}, \mathcal{T}^1, \tau_l^1, \mathcal{T}^i, \tau_l^i, ...]$ | Indicates all input tokens of LLM |
| $\mathcal{T}_{task}$ | Task text prompt tokens |
| $\tau_l^i$ | Task target organ tokens |
| $\mathcal{T}^i. i \in [1, N]$ | Represents the N potential related or ROI organs/suborgans reasoned by LLM. |
| $\mathcal{R}_{cot}^1$ | The initial diagnostic result output by the LLM |
| $M_l^1$ | Local ROI mask |
| $L_{total}$ | Training loss |
| $N$ | Represents the output text length |
| $M$ | represents the sample number |
| $\alpha$ | Loss weight |

## C RELATED WORK

**Medical Benchmark.** Early public medical benchmarks such as VQA-RAD (Lau et al., 2018), VQA-Med (Ben Abacha et al., 2019), SLAKE (Liu et al., 2021), and PathVQA (He et al., 2020) have greatly advanced LVLM development in healthcare. These datasets are primarily 2D and feature simple, template-based QA pairs. Recent works employ data synthesis to build larger and more multimodal benchmarks: RadFM (Wu et al., 2023) generates expert-verified questions from literature and cases. OmniMedVQA (Hu et al., 2024) leverages category-driven templates with GPT-4 (Achiam et al., 2023) for paraphrasing and distractor creation. HealthBench (Arora et al., 2025) combines synthetic generation and adversarial testing for multi-turn, multilingual dialogues. In the field of tumor analysis, DeepTumorVQA (Chen et al., 2025a) focuses on fine-grained 3D CT tumor detection. However, most benchmarks remain limited to multiple-choice formats and basic reasoning, insufficient to meet clinical demands for interpretability and in-depth analysis.

**Medical Large Vision-Language Models.** Med-LVLMs such as LLaVA-Med (Li et al., 2023), Lingshu (Xu et al., 2025b), Med-PaLM (Singhal et al., 2025) and HealthGPT (Lin et al., 2025) have rapidly advanced multimodal clinical decision support by aligning ViT-based (Zhou et al., 2024) 2D image features with language models. Methods like M3D (Bai et al., 2024), CT-Chat (Hamamci et al., 2024a) and Merlin (Blankemeier et al., 2024) incorporate 3D CNNs and ViTs to better capture volumetric data, enabling more precise analysis of clinical 3D images. However, these Med-LVLMs lack causal reasoning for complex cases, prompting a shift toward CoT and knowledge-guided reasoning paradigms. MedVLThinker (Huang et al., 2025) and MedVLM-R1 (Pan et al., 2025) combine CoT reasoning with reinforcement learning to improve interpretability, while ReasonMed (Sun et al., 2025) optimizes reasoning paths via multi-agent verification. MedResearcher-R1 (Yu et al., 2025a) enhances deep reasoning by leveraging knowledge graphs for multi-hop QA sample construction. By emphasizing logical chains, medical vision reasoning models are overcoming traditional 2D and 3D VLM limitations and opening new directions for explainable and reliable clinical decision-making.

**Online Interleaved Reasoning.** Online interleaved reasoning (IR) dynamically alternates thinking and answering, enabling more efficient multi-hop reasoning than traditional methods (Qiu et al., 2020; Trivedi et al., 2022). IRCoT (Trivedi et al., 2022) shows that interleaving retrieval, reasoning, and response generation significantly improves performance on complex tasks. Chain-of-Focus (Zhang et al., 2025) further introduces adaptive visual search and zooming with reinforcement learning to enhance multimodal reasoning. CX-Mind (Li et al., 2025) adopts interleaved reasoning strategies for chest X-ray diagnosis, generating verifiable reasoning chains. Online IR aligns Med-LVLM decision-making with clinical workflows, motivating us to explore IR for 3D tumor analysis.

# D    EXPERIMENTS AND IMPLEMENTATION DETAILS

## D.1    TRAINING DETAILS

**Data Preparation and Preprocessing.** TumorChain takes as input complete three-dimensional CT scans in `.nii.gz` format. It is important to note that these volumetric images exhibit considerable variability in spatial dimensions along the x, y, and z axes. To ensure consistency and model stability, we employ a multi-step preprocessing pipeline.

Initially, we extract the soft tissue window level and window width for each CT scan according to standard radiological protocols, followed by normalization of voxel intensities. This step reduces inter-scan variability and facilitates robust feature extraction. Given the heterogeneity in image shapes, we address dimensional inconsistency by applying precise cropping and zero-padding strategies. These operations are carefully performed to prevent geometric distortion and loss of anatomical information.Finally, all processed CT volumes are resized to a fixed shape of (256, 256, 32)(height × width × depth), in order to meet the input requirements of the 3D vision encoder. This standardized preprocessing framework ensures that all input volumes possess the same spatial resolution, thereby improving model compatibility and computational efficiency.

**Model Architecture.** We adopt Qwen2.5-VL-3B and Qwen2.5-VL-7B as the base multimodal backbones for TumorChain-3B and TumorChain-7B, respectively. Since the original Qwen-VL vision encoder only supports 2D images, we replace it with M3D encoder (Bai et al., 2024) to support volumetric 3D CT inputs. To bridge visual and language modalities, we employ a two-layer multilayer perceptron (MLP) projection module, consisting of two Linear layers with an intermediate ReLU activation, to map the extracted organ-level visual embeddings into the LLM token space and facilitate multimodal fusion. For the large language model component, the pretrained LLM backbone of Qwen2.5-VL-3B/7B is used for high-level multimodal reasoning and report generation. Additionally, an auxiliary classification head is implemented as a single fully connected layer to perform binary classification (normal vs. abnormal) on each segmented organ region, providing supervisory calibration. Both TumorChain models are trained under identical hyperparameter configurations, including learning rate and precision settings, following the hybrid model fine-tuning strategy described in the main text.

**Hardware and Distributed Training.** Model training is conducted on 32 NVIDIA A800 GPUs. For the 3B model, training is performed for 12 hours, whereas the 7B model is trained for 16 hours. The training set comprises 1.5M CT samples, utilizing 90% of the available data. All models are

Figure 7: Semantic Consistency Evaluation Prompt.

trained using the DeepSpeed distributed framework (configuration: ZeRO-2), effectively enabling large-scale parallel training and efficient memory utilization.

**Hyperparameter Configuration.** All training and evaluation pipelines are conducted with strict reproducibility control. Detailed hyperparameter settings are as shown in table 5.

## D.2 SEMANTIC ACCURACY METRIC BY GPT5

To assess the quality of open-ended answers in the VQA task for CT imaging, we introduce a semantic consistency metric specifically designed for shape, boundary, density, count, TNM stage prediction, and other clinical questions. For each question–answer pair, the evaluator first classifies the question type, identifies its key clinical focus, and then compares the model's answer to the ground truth (reference answer). The metric judges the response as correct if the essential medical meaning and main clinical finding are preserved, regardless of minor differences in expression or reasoning process. This approach enables reliable evaluation of the model's clinical interpretability and factual accuracy, focusing on whether the core diagnostic content is communicated consistently. The semantic consistency metric ensures that the assessment is robust against linguistic variation and prioritizes agreement in clinical understanding over superficial textual similarity.

An illustrative example of the evaluation prompt used to guide semantic consistency assessment is shown in Figure 7.

## D.3 EXPERIMENT RESULT DETAILS

### D.3.1 EXPERIMENTAL RESULTS BY ORGAN CATEGORY

In this section, we present more detailed ablation studies and comprehensive Chain-of-Thought (CoT) evaluation results than those included in the main text. Table 6 and Table 7 summarize the organ-level accuracy of our model and various baselines on the TumorCoT and DeepTumorVQA benchmarks, respectively. The scope of our evaluation focuses on the five major digestive organs: liver, stomach, pancreas, colon, and esophagus. For generalization tests, we specifically analyze the overlapping three organs—liver, pancreas, and colon—within the DeepTumorVQA dataset, as these closely match our benchmark task settings.

Table 5: Hyperparameter Configuration

| Hyperparameter | Value |
| --- | --- |
| Optimizer | AdamW |
| Learning rate | $3 \times 10^{-5}$ |
| Learning rate scheduler | cosine decay |
| Weight decay | 0.0 |
| Warmup ratio | 0.03 |
| Mixed precision | bf16 (fp16 disabled) |
| Gradient accumulation steps | 2 |
| Per-device batch size | 2 |
| Number of dataloader workers | 16 |
| Number of epochs | 1.0 |

Table 6: CoT-Report comparison of TumorChain with other **Reasoning VLMs** on `TumorCoT`.

| Reasoning Models | Liver | Pancreas | Colon | Stomach | Esophagus | Average |
|---|---|---|---|---|---|---|
| Claude3-Haiku | 34.70 | 56.13 | 44.08 | 17.11 | 39.77 | 38.36 |
| Gemini2.0-Flash | 42.35 | 29.03 | 56.24 | 71.15 | 38.64 | 47.48 |
| GPT-5-Mini | 44.48 | 35.71 | 59.76 | 76.85 | 65.12 | 56.38 |
| Qwen2.5-VL-7B | 47.94 | 58.15 | 54.60 | 69.79 | 52.41 | 56.58 |
| InternVL-2.5-8B | 48.89 | 49.20 | 48.89 | 55.84 | 52.54 | 51.07 |
| MiniCPM-V-4.5-9B | 46.07 | 43.29 | 45.69 | 50.26 | 47.59 | 46.58 |
| HealthGPT-3.8B | 48.69 | 63.74 | 36.44 | 31.74 | 45.81 | 45.28 |
| Lingshu-7B | 53.03 | 33.55 | 53.23 | 60.71 | 44.42 | 48.99 |
| MedVLM-R1-2B | 38.58 | 42.78 | 37.18 | 54.79 | 34.90 | 41.65 |
| RadFM | 11.34 | 15.42 | 2.50 | 35.20 | 13.71 | 15.63 |
| M3D-Phi3-4B | 21.51 | 28.76 | 9.51 | 52.57 | 17.72 | 26.01 |
| **TumorChain-3B** | 69.89 | 72.20 | 67.39 | 91.63 | 64.34 | 73.09 |
| **TumorChain-7B** | **83.45** | **78.43** | **68.65** | **93.45** | **88.71** | **82.54** |

Table 7: Comparison of TumorChain with other VLMs on the public DeepTumorVQA benchmark.

| Model | Liver | Pancreas | Colon | Average |
|---|---|---|---|---|
| *Generalist Models* | | | | |
| Qwen2.5-VL-7B | 36.32 | 32.39 | 46.33 | 38.35 |
| InternVL-2.5-8B | 34.52 | 35.86 | 52.75 | 41.04 |
| MiniCPM-V4.5-9B | 40.02 | 45.20 | 52.93 | 46.05 |
| *Commercial Models* | | | | |
| Claude3-Haiku | 32.67 | 32.62 | 18.40 | 27.90 |
| Gemini2.0-Flash | 35.33 | 35.02 | 44.80 | 38.38 |
| GPT-5-mini | 41.20 | 42.48 | 73.60 | 52.43 |
| *2D Medical Models* | | | | |
| HealthGPT-3.8B | 30.32 | 30.26 | 53.56 | 38.05 |
| Lingshu-7B | 38.39 | 38.01 | 50.68 | 42.36 |
| MedVLM-R1-2B | 35.98 | 36.73 | 69.85 | 47.52 |
| *3D Medical Models* | | | | |
| RadFM | 31.97 | 34.11 | 55.71 | 40.60 |
| M3D-Phi3-4B | 24.82 | 24.88 | 19.68 | 23.13 |
| **TumorChain-7B** | **55.41** | **62.45** | **84.33** | **67.40** |

### D.3.2 ABLATION EXPERIMENTS UNDER DIFFERENT CLASSIFICATION LOSS WEIGHT

Table 8 reports the results of ablation experiments under different classification loss weight ($\alpha$) settings. We systematically investigate the impact of varying $\alpha$, and the findings show that moderately increasing the classification loss weight (e.g., $\alpha$=1.0) yields significant performance improvements across multiple tasks, with a highest average accuracy of 84.41%. In contrast, setting $\alpha$ either too low or too high leads to reduced performance. These results highlight the critical role of appropriate loss weight adjustment in optimizing the collaborative training of hybrid models.

### D.3.3 ABLATION EXPERIMENTS ON BASELINES BEFORE AND AFTER FINE-TUNING

Our contributions include both a large-scale, high-quality dataset and a novel model architecture. Accordingly, our experiments cover three aspects: main evaluations demonstrating overall performance, ablation studies assessing the impact of individual model components, and generalization tests exploring robustness across different data distributions. The main results in the paper demonstrate significant improvements across multiple metrics. In addition, we conducted rigorous ablation studies on three key components of the architecture—Impression-Information Retrieval (IIR), classification-loss weighting, and CoT-formatted data. The results indicate that each module delivers clinically meaningful performance gains.

To further isolate and validate the contribution of the dataset itself, we fine-tuned three representative baseline models on our training set to facilitate direct performance comparisons and thoroughly assess the gains attributable to the dataset alone. The supplemental results, shown as Table 9, reveal

Table 8: Ablation study of the effect of classfication loss weight $\alpha$ in TumorChain.

| $\alpha$ | Position | | Lesion Attributes | | | | | | TNM Prediction | | | CoT-Report | Avg. |
|---|---|---|---|---|---|---|---|---|---|---|---|---|---|
| | Organ Pos. | Tumor Pos. | Seg. Loc. | Shape | Boundary | Density | Count | Others | Tumor | Node | Met. | | |
| **0.0** | 99.92 | 97.01 | 82.46 | 81.45 | 75.82 | 83.01 | 80.89 | 82.35 | 86.11 | 57.88 | 65.22 | 77.41 | 80.79 |
| **0.1** | 99.90 | 95.58 | 84.01 | 79.78 | 75.09 | 82.10 | 83.85 | 81.85 | 88.69 | 58.16 | 69.54 | 78.29 | 81.40 |
| **0.5** | 99.90 | 97.14 | 83.14 | 81.60 | 81.15 | 83.51 | 84.01 | 82.45 | 86.43 | 60.85 | 70.65 | 79.54 | 82.53 |
| **1.0** | 99.97 | 97.57 | 86.88 | 82.28 | 84.52 | 85.05 | 86.20 | 86.57 | 88.83 | 61.63 | 71.07 | 82.36 | 84.41 |
| **1.5** | 99.94 | 97.91 | 83.21 | 79.40 | 74.46 | 75.37 | 81.83 | 77.41 | 86.23 | 57.51 | 57.46 | 79.21 | 79.16 |

Table 9: Ablation study of baselines before and after finetuning on TumorCoT.

| | Avg. Score (Before finetuning / After finetuning) | | | | |
|---|---|---|---|---|---|
| **Model** | **Position** | **Lesion Attributes** | **TNM Prediction** | **CoT Report** | **Avg.** |
| M3D-Phi-3-4B | 30.36 / **86.92** | 33.23 / **61.78** | 37.16 / **59.48** | 34.79 / **61.18** | 32.84 / **66.98** |
| Lingshu-7B | 28.93 / **85.46** | 49.91 / **77.36** | 43.21 / **67.65** | 54.30 / **65.98** | 45.10 / **74.11** |
| Qwen2.5-VL-7B | 31.05 / **91.02** | 47.17 / **76.45** | 40.94 / **62.37** | 60.54 / **74.21** | 44.04 / **76.01** |
| TumorChain-7B | - / **98.77** | - / **85.25** | - / **73.84** | - / **82.36** | - / **84.41** |

that all baseline models achieve notable improvements after fine-tuning, underscoring the unique and substantial value of our dataset within the tumor domain.

Meanwhile, our proposed TumorChain-7B continues to significantly outperform these fine-tuned baselines, reflecting the effectiveness of our architectural innovations.

# E INTERACTIVE-VALIDATED COT DATA ENGINE

## E.1 DATA ENGINE COMPONENTS DETAILS

**(i) Diagnostic Knowledge Graph.** In collaboration with five organ-specialist radiologists, we construct a triplet-based ("entity–relation–entity") knowledge graph from diagnostic guidelines, textbooks, and representative cases, covering anatomy, findings, impressions, histopathology, and risk factors (see Appendix Figure 8). All organ segmentation standards follow international conventions, enabling hierarchical substructure and tumor-grade analysis. During reasoning chain construction, the data agent retrieves relevant nodes and relations from the knowledge graph to ensure traceable and reliable logic and minimize factual and logical errors.

**(ii) Structured Feature Extractor.** We employ Qwen3-235B-A22B as the structured feature extractor to perform rigorous cleaning and terminology standardization across multi-source, multi-format data, with all imaging descriptors aligned to the RadLex international standard. Leveraging organ segmentation from the knowledge graph, the module automatically extracts substructure-level descriptions and structured features—including clinical information (age, gender), lesion attributes (location, shape, margin, density, count), and six additional categories—from CT and pathology reports. TNM staging information and final diagnostic conclusions are also extracted from pathology reports, ensuring the completeness and accuracy of subsequent reasoning chains.

**(iii) Segmentation Expert Model.** We utilize TotalSegmentator to segment 117 organs and merge them to 56 organs (see Appendix Table 10). The organ and tumor masks of five digestive organs are refined by radiologists, enabling structured ROI localization for subsequent VQA tasks.

**(iv) Traceable CoT Reasoner.** GPT-4o-mini, selected for strong language and medical expertise, receives multidimensional structured reports and knowledge graph links, generating organ- and lesion-level VQA that conforms to medical guidelines.

**(v) CoT Logic Calibrator.** During reasoning chain generation, Claude3.5-Haiku automatically verifies logical integrity. Upon detecting inconsistencies, the system uses two prompt strategies: expanding organ regions via segmentation for re-reasoning, or requesting further clarification and self-reflection from the reasoning module.

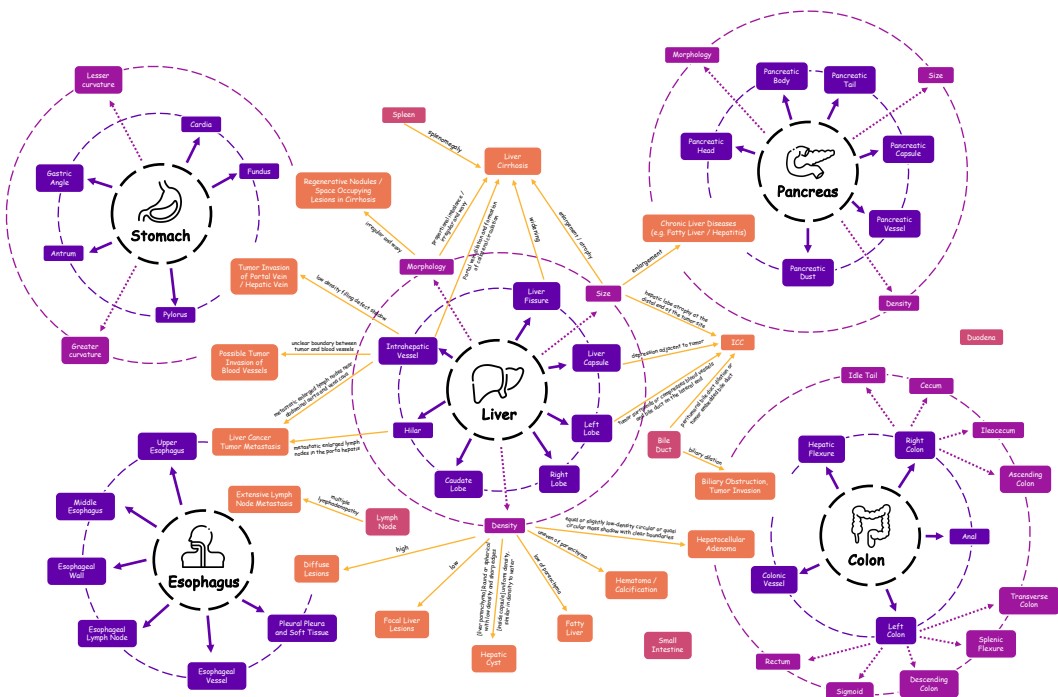

Figure 8: The overview of the diagnostic knowledge graph of 5 organs by doctors.

**(vi) Summarizer for Finding-Impression-Pathology Chains.** We employ GPT-5-mini to aggregate all reasoning chains. This module receives structured information from pathology reports to construct a higher-level pathology analysis VQA. If inconsistencies arise between reasoning chains and pathological conclusions, the system automatically reinitiates inference. Upon successful validation, it extracts findings and impressions related to staging, lymph node involvement, and metastasis, generating TNM prediction VQA pairs. Finally, a traceable CoT-format report is produced for each organ, facilitating standardized and interpretable reasoning outputs.

### E.2 DIAGNOSTIC KNOWLEDGE GRAPH

As figure 8 shows, the diagnostic knowledge graph constructed in this study serves as the core clinical prior for the TumorCoT-1.5M dataset and TumorChain model, providing structured, traceable medical knowledge support for multimodal CoT reasoning in tumor analysis. Unlike general medical knowledge graphs that cover broad healthcare domains, this KG is tumor-centric and organ-specific, focusing exclusively on the five major digestive organs (esophagus, stomach, colon, pancreas, and liver) closely associated with clinical tumor diagnosis and treatment. Its core value lies in bridging the gap between unstructured medical text (e.g., radiology reports) and structured reasoning logic, ensuring that every step of the CoT reasoning process (from imaging findings to pathology predictions) is grounded in evidence-based medical knowledge rather than arbitrary inference.

The construction of the KG adheres to a multi-source, hierarchical verification principle, integrating three levels of authoritative data sources to balance comprehensiveness, accuracy, and clinical relevance. **A-level** primarily includes international and domestic clinical diagnostic guidelines of five organs, which define the core knowledge boundaries of the KG, including TNM staging criteria and standard imaging features of malignant tumors. **B-level** covers classic textbooks and high-impact academic papers. These sources supplement critical anatomical details and advanced imaging features. **C-level** consists of real-world expert-annotated cases, including typical tumor cases labeled by 5 experienced organ-specialist radiologists and teaching cases from the Department of Radiology at top-tier tertiary hospitals.

This design not only ensures that the knowledge conforms to international clinical guidelines but also incorporates real-world diagnostic experience from expert physicians, making it more suitable for high-stakes tumor analysis scenarios.

```
You are a professional radiologist and CT imaging doctor. I am building a CT imaging diagnostic model and need a VQA dataset for training.
Input: CT imaging findings related to the liver from a clinical case report: <FINDINGS>
Your Task: Based on these findings, construct the following 6 categories of independent VQA pairs (no multi-turn dialogue, no information linkage):
1. Segmental localization: Construct VQA pairs on segmental localization if the given report sentences mentions tumor or space-occupying lesions with
locations. The answer must be concise, mentioning liver segments including the left lobe, right lobe, and caudate lobe.
     Example:
     Original description: "A mixed slightly hypodense lesion about 2.7 × 2.3 cm was found in S4b of the left lobe of the liver."
     Question: "In which segment of the liver is the lesion located?"
     Answer:"Reasoning: When examining this CT image, I look for any abnormal density within the liver parenchyma. In the left lobe, specifically in segment
     S4b, I observe a mixed slightly hypodense lesion measuring about 2.7 × 2.3 cm. Therefore, I infer that the lesion is in the left lobe of the liver.
     Summary: Thus, the answer is left lobe."
2. Shape: Construct VQA pairs from sentences if they include descriptions of size, shape, thickness (thickened), dilation, stenosis, or the shape of a lesion/
low-density shadow.
     Example:
     Original description: "The distal part of the middle hepatic vein is poorly demarcated from the mass, with localized luminal narrowing."
     Question: "Is there evidence of luminal narrowing in the middle hepatic vein?"
     Answer: "Reasoning: The middle hepatic vein (MHV) runs between the left and right lobes of the liver (near the Cantlie line) and drains into the inferior
     vena cava. By examining the CT image, I locate the middle hepatic vein and observe that its lumen is irregular, with localized narrowing distally.
     Summary: Thus, the answer is that the middle hepatic vein shows localized narrowing."
3. Boundary: Construct boundary-related VQA pairs if there are descriptions of boundary (clear/unclear) or margin (smooth/intact), such as whether the organ
margins are smooth and complete or whether inter-organ boundaries are distinct.
     Example:
     Original description: "The margins of the pancreas appear smooth and well defined, with a clear boundary from the surrounding fat tissue."
     Question: "Are the margins of the pancreas smooth and clearly demarcated from surrounding tissues?"
     Answer: "Reasoning: When evaluating the pancreas on CT imaging, I check whether its edges are intact and whether the transition to adjacent fat tissue is
     distinct. In this case, the margins are described as smooth and well defined, with a clear separation from the surrounding fat tissue. This indicates that
     the pancreatic boundary is intact and distinguishable. Summary: Thus, the answer is that the pancreatic margins are smooth and clearly demarcated."
4. Density: Construct VQA pairs of Density if there are descriptions related to density, such as organ density, or hypodense/hyperdense lesions in
substructures.
     Example:
     Original description: "A hypodense lesion is observed in the pancreatic head, measuring approximately 2.5 cm."
     Question: "What is the density characteristic of the lesion in the pancreatic head?"
     Answer: "Reasoning: On CT imaging, density changes are key to characterizing lesions. In the pancreatic head, I detect an abnormal lesion that appears
     darker than the surrounding pancreatic parenchyma, indicating reduced attenuation. This is described as hypodense. Summary: Thus, the answer is
     hypodense."
5. Counting: Construct diverse counting-related questions if the report clearly states the number of lesions in a liver segment (e.g., how many? solitary?
multiple?).
     Example:
     Original description: "Two hypodense lesions, one in segment S6 and another in S7 of the right lobe of the liver, are observed on CT."
     Question: "How many hypodense lesions are present in the liver?"
     Answer: "Reasoning: When reviewing the CT scan of the liver, I note two distinct hypodense lesions. One is located in segment S6 and the other in segment
     S7 of the right lobe. These represent two separate lesions, not a single confluent abnormality. Summary: Thus, the answer is 2."
6. Others: Construct relevant questions if there are other key imaging findings not covered by the above categories.
Output Rules:
· Each answer must include a chain-of-thought (reasoning) process.
· Extract key information from the provided findings and construct questions and reasoning chains, without mentioning or referencing the original report.
· Use diverse expressions and sentence structures.
· The summary format of each answer should be: "Thus, the answer is …".
· In reasoning, apply appropriate medical imaging knowledge.
· Avoid redundancy; prioritize important findings.
```

Figure 9: Prompt example: Traceable CoT Reasoner for extracting lesion attributes.

### E.3    PROMPT DESIGN OF DATA ENGINE

When designing prompt templates for each agent in our data engine, we adhere to the following principles to ensure both data quality and clinical validity.

**Standardization of Medical Terminology.** All descriptions are required to follow internationally recognized medical terminology standards (e.g., RadLex Radiological Lexicon, AJCC TNM staging system). This prevents ambiguity and ensures consistency across tasks (for example, using "hypodense lesion" instead of the vague expression "low-density shadow").

**Mandatory Reasoning Chain.** Each response must consist of two components: a "Reasoning Process" and a "Summary." The reasoning process should explicitly reflect clinical diagnostic logic, typically progressing from organ localization to feature observation and finally to pathological correlation. The summary should present the diagnostic conclusion in a standardized form, e.g., "Thus, the answer is ...".

**Task Boundary Specification.** To avoid cross-task redundancy, prompts are designed to clearly delineate the input scope for different subtasks (e.g., lesion localization tasks focus exclusively on anatomical positioning without involving density assessment or malignancy determination).

**Multi-Source Information Integration.** Prompts are required to guide the agent to incorporate both structured medical knowledge (e.g., anatomical ontologies, tumor metastasis pathways) and quantitative image-derived features (e.g., lesion boundary, density values). This ensures that the reasoning process is grounded in objective evidence rather than subjective speculation.

Following these principles, the CoT inference framework aligns with clinical diagnostic standards and effectively supports the four core tasks of tumor analysis: anatomical localization, lesion attribute characterization, TNM staging prediction, and structured report generation. An illustrative example of a prompt designed for the Traceable CoT Reasoner to extract lesion attributes is shown in Figure 9.

### E.4 MULTI-ORGAN MASKS AND MERGED ORGAN MASKS

Table 10 provides the detailed organ mask IDs from TotalSegmentor (Wasserthal et al., 2023) and the mapping with our merged organ mask IDs.

Table 10: Mapping Table of TotalSegmentator Organ Indexes After Aggregation.

| New Label | Segmentator Name | Included Organs | TotalSeg. Label |
|---|---|---|---|
| 0 | background | background | 0 |
| 1 | spleen | spleen | 1 |
| 2 | kidney_right | kidney_right | 2 |
| 3 | kidney_left | kidney_left | 3 |
| 4 | gallbladder | gallbladder | 4 |
| 5 | liver | liver | 5 |
| 6 | stomach | stomach | 6 |
| 7 | pancreas | pancreas | 7 |
| 8 | adrenal_gland_right | adrenal_gland_right | 8 |
| 9 | adrenal_gland_left | adrenal_gland_left | 9 |
| 10 | lung_upper_lobe_left | lung_upper_lobe_left | 10 |
| 11 | lung_lower_lobe_left | lung_lower_lobe_left | 11 |
| 12 | lung_upper_lobe_right | lung_upper_lobe_right | 12 |
| 13 | lung_middle_lobe_right | lung_middle_lobe_right | 13 |
| 14 | lung_lower_lobe_right | lung_lower_lobe_right | 14 |
| 15 | esophagus | esophagus | 15 |
| 16 | trachea | trachea | 16 |
| 17 | thyroid_gland | thyroid_gland | 17 |
| 18 | small_bowel | small_bowel | 18 |
| 19 | duodenum | duodenum | 19 |
| 20 | colorectum | colon | 20 |
| 21 | urinary_bladder | urinary bladder | 21 |
| 22 | prostate | prostate | 22 |
| 23 | kidney_cyst | kidney_cyst_left | 23 |
|  |  | kidney_cyst_right | 24 |
| 24 | vertebrae_S1 | vertebrae_S1 | 26 |
| 25 | lumbar_vertebrae | vertebrae_L5 | 27 |
|  |  | vertebrae_L4 | 28 |
|  |  | vertebrae_L3 | 29 |
|  |  | vertebrae_L2 | 30 |
|  |  | vertebrae_L1 | 31 |
| 26 | thoracic_vertebrae | vertebrae_T12 | 32 |
|  |  | vertebrae_T11 | 33 |
|  |  | vertebrae_T10 | 34 |
|  |  | vertebrae_T9 | 35 |
|  |  | vertebrae_T8 | 36 |
|  |  | vertebrae_T7 | 37 |
|  |  | vertebrae_T6 | 38 |
|  |  | vertebrae_T5 | 39 |
|  |  | vertebrae_T4 | 40 |
|  |  | vertebrae_T3 | 41 |
|  |  | vertebrae_T2 | 42 |
|  |  | vertebrae_T1 | 43 |
| 27 | cervical_vertebrae | vertebrae_C7 | 44 |
|  |  | vertebrae_C6 | 45 |
|  |  | vertebrae_C5 | 46 |
|  |  | vertebrae_C4 | 47 |
|  |  | vertebrae_C3 | 48 |
|  |  | vertebrae_C2 | 49 |
|  |  | vertebrae_C1 | 50 |
| 28 | sacrum | sacrum | 25 |
| 29 | humerus | humerus_left | 69 |
|  |  | humerus_right | 70 |
| 30 | scapula | scapula_left | 71 |
|  |  | scapula_right | 72 |
| 31 | clavicula | clavicula_left | 73 |
|  |  | clavicula_right | 74 |
| 32 | femur | femur_left | 75 |
|  |  | femur_right | 76 |
| 33 | hip | hip_left | 77 |
|  |  | hip_right | 78 |
| 34 | rib_left | rib_left_1 | 92 |
|  |  | rib_left_2 | 93 |

| | | rib_left_3 | 94 |
|---|---|---|---|
| | | rib_left_4 | 95 |
| | | rib_left_5 | 96 |
| | | rib_left_6 | 97 |
| | | rib_left_7 | 98 |
| | | rib_left_8 | 99 |
| | | rib_left_9 | 100 |
| | | rib_left_10 | 101 |
| | | rib_left_11 | 102 |
| | | rib_left_12 | 103 |
| 35 | rib_right | rib_right_1 | 104 |
| | | rib_right_2 | 105 |
| | | rib_right_3 | 106 |
| | | rib_right_4 | 107 |
| | | rib_right_5 | 108 |
| | | rib_right_6 | 109 |
| | | rib_right_7 | 110 |
| | | rib_right_8 | 111 |
| | | rib_right_9 | 112 |
| | | rib_right_10 | 113 |
| | | rib_right_11 | 114 |
| | | rib_right_12 | 115 |
| 36 | sternum | sternum | 116 |
| 37 | costal_cartilages | costal_cartilages | 117 |
| 38 | heart | heart | 51 |
| 39 | aorta | aorta | 52 |
| 40 | pulmonary_vein | pulmonary_vein | 53 |
| 41 | brachiocephalic_trunk | brachiocephalic_trunk | 54 |
| 42 | subclavian_artery | subclavian_artery_right | 55 |
| | | subclavian_artery_left | 56 |
| 43 | common_carotid_artery | common_carotid_artery_right | 57 |
| | | common_carotid_artery_left | 58 |
| 44 | brachiocephalic_vein | brachiocephalic_vein_left | 59 |
| | | brachiocephalic_vein_right | 60 |
| 45 | atrial_appendage_left | atrial_appendage_left | 61 |
| 46 | superior_vena_cava | superior_vena_cava | 62 |
| 47 | inferior_vena_cava | inferior_vena_cava | 63 |
| 48 | portal_vein_and_splenic_vein | portal_vein_and_splenic_vein | 64 |
| 49 | iliac_artery | iliac_artery_left | 65 |
| | | iliac_artery_right | 66 |
| 50 | iliac_vena | iliac_vena_left | 67 |
| | | iliac_vena_right | 68 |
| 51 | gluteus | gluteus_maximus_left | 80 |
| | | gluteus_maximus_right | 81 |
| | | gluteus_medius_left | 82 |
| | | gluteus_medius_right | 83 |
| | | gluteus_minimus_left | 84 |
| | | gluteus_minimus_right | 85 |
| 52 | autochthon | autochthon_left | 86 |
| | | autochthon_right | 87 |
| 53 | iliopsoas | iliopsoas_left | 88 |
| | | iliopsoas_right | 89 |
| 54 | spinal_cord | spinal_cord | 79 |
| 55 | brain | brain | 90 |
| 56 | skull | skull | 91 |

### E.5 QUANTITATIVE EVALUATION OF THE COT DATA ENGINE BY EXPERT CLINICIANS.

To systematically assess the factual correctness and clinical plausibility of the generated reasoning chains, we randomly sampled 5,000 VQA instances from the TumorChain-1.5M dataset across task types and submitted them to a team led by a board-certified radiologist(8 years of oncologic imaging experience) for evaluation from two perspectives:

• **Usability**: The clinician first assessed whether the VQA result matches the corresponding information in the original medical report. If the VQA result is correct and can provide reliable support in clinical practice, it is marked as usable; otherwise, it is marked as unusable.

• **Clinical Reasonableness**: For all usable data, we further categorized the samples:

a. High quality: For usable cases, the reasoning chain must exhibit completeness, scientific soundness, and logical coherence; only when all criteria are satisfied is it marked as high quality.

b. Acceptable: For usable cases where the overall logic is sound but some steps are slightly brief or marginally insufficient, it is marked as acceptable.

Evaluation results show that the usability rate reaches 95.88%, and among all usable samples, 97.85% are high-quality. These results validate the effectiveness of our data-generation pipeline and demonstrate the reliability and practical clinical value of the dataset.

## F  CoT Evaluation Details: TumorChain-Eval

This appendix provides a comprehensive description of the evaluation framework for Chain-of-Thought (CoT) reasoning. The framework of **TumorChain-Eval** is organized into two stages. In the first stage, structured (subject, relation, object) triples are extracted from medical texts to serve as factual building blocks for reasoning. In the second stage, the generated CoT reasoning chains are assessed for quality based on these triples.

### F.1  Stage 1: Extraction of Medical Entity–Relation-Entity Triples

This stage serves as a prerequisite for CoT evaluation. Its goal is to convert unstructured medical text into structured knowledge representations, thereby supplying accurate and unambiguous factual units for subsequent reasoning-chain assessment.

**Task Definition.** Given a medical radiology report, the objective is to extract all (subject, relation, object) triples that capture medical facts. *Subject*: Typically an organ (e.g., liver), an anatomical structure (e.g., pericolonic vessels), or a lesion (e.g., enlarged lymph node). *Relation*: A verb or state term that describes the medical semantic relationship between the subject and the object (e.g., not observed, suggests, absent, supports). *Object*: Generally a medical finding, lesion (e.g., wall thickening, abnormal density), condition (e.g., mass-like change), or another related organ.

Table 11: Prompts for medical triples extraction in stage 1 of CoT evaluation

| **Prompt for Medical Fact Triple Extraction** |
| --- |
| You are a medical fact–extraction assistant specialized in parsing knowledge from the given clinical text. Your task is to analyze the input and extract all factual triples $(subject, relation, object)$, where each triple expresses a single, clear medical fact. Ensure that all information is presented strictly in triple form, focusing on **organs/structures**, **relations**, and **lesions**. |
| **Input Example:** First, examine the right hemicolon, cecum, ascending colon, and hepatic flexure; no wall thickening or abnormal density is observed, indicating no definite mass or inflammatory change in the right hemicolon. Second, check the left hemicolon, transverse colon, and splenic flexure, descending colon, sigmoid colon, and rectum; again no abnormal thickening or abnormal density is found, supporting the absence of significant lesions in the left hemicolon. Third, evaluate colon-related internal structures and surrounding tissues; no pericolonic vascular abnormality is detected, and the liver, gallbladder, pancreas, spleen, kidneys, and pelvic structures show no colon-related significant lesion, enlarged lymph nodes, or free fluid. These findings collectively do not support tumor or active inflammatory changes. Overall, the imaging features do not suggest colorectal cancer or other qualifying pathology. |
| **Output Requirements:** In each triple, the first and third elements must be an organ or lesion, and the second element must describe their relationship. If the second and third positions are reversed, correct them. The output format should be: 

 (Right hemicolon, not observed, wall thickening); |

**Prompt Design and Specification** To guide the model in performing triple extraction, we design standardized prompts tailored to the characteristics of medical tasks. The prompt clearly specifies

Table 12: Finding Chain (FC) scoring criteria.

| Chain Level | Scoring Dimension | Description | Scoring Criteria |
|---|---|---|---|
| Finding Chain | Existence Match | Degree to which predicted facts match the ground-truth (GT) facts. | **10**: Prediction contains all key GT facts with no omission or redundancy. |
| | | | **8–9**: Prediction covers the vast majority of GT facts, with only minor omissions or redundancies (¡10%). |
| | | | **6–7**: Prediction covers some GT facts but exhibits moderate omissions or redundancies (10–30%). |
| | | | **4–5**: Prediction covers only a small portion of GT facts; substantial omissions or redundancies (30–50%). |
| | | | **1–3**: Low match rate; only a few facts are correct and the error rate is very high. |
| | | | **0**: Prediction includes none of the GT facts. |
| | Completeness | Extent to which key facts are fully expressed without missing or spurious content. | **10**: Prediction covers 100% of GT facts with no omissions. |
| | | | **8–9**: Only minor omissions ($<10\%$), high coverage. |
| | | | **6–7**: Noticeable omissions (10–30%), coverage is moderately compromised. |
| | | | **4–5**: Majority of key facts are missing (30–50%), poor coverage. |
| | | | **1–3**: Extensive omissions ($>50\%$), very low coverage. |
| | | | **0**: Prediction fails to cover any key facts. |
| | Accuracy | Correctness of factual statements in the prediction. | **10**: All predicted entries are accurate with no invalid or incorrect statements. |
| | | | **8–9**: Vast majority accurate ($<10\%$ problematic), only minor issues. |
| | | | **6–7**: Moderate errors or invalid entries (10–30%) that negatively affect overall quality. |
| | | | **4–5**: Numerous erroneous entries ($>30\%$), only a small fraction correct. |
| | | | **1–3**: Very few accurate entries ($>50\%$ error rate). |
| | | | **0**: All predictions are invalid; no correct entries. |

the model's role, the extraction objective, and the required output format, while providing illustrative examples to facilitate structured responses. The prompt for LLM to extract medical entity-relation-entity triples is shown in Table 11.

## F.2 STAGE 2: CoT EVALUATION PROTOCOL

Following the triple-extraction stage, this phase aims to comprehensively assess the quality of the model-generated Chain-of-Thought (CoT) reasoning chains. Evaluation is performed by comparing model predictions (Pred) against the reference ground truth (GT).

**Reasoning-Chain Categorization.** Both the predicted (Pred) and reference (GT) reasoning chains are segmented into three hierarchical levels to enable fine-grained evaluation: *(i) Finding Chain (FC):* Consists of basic facts directly extracted from radiological descriptions (e.g., no wall thickening observed, soft-tissue mass detected). These represent independent, objective observations without inferential steps. *(ii) Impression Chain (IC):* Comprises intermediate medical impressions or preliminary suggestions derived from multiple S1 findings (e.g., suggests localized inflammatory change). This level reflects simple, initial reasoning. *(iii) Long Reasoning Chain (LRC):* Represents high-level medical reasoning and conclusions that integrate all findings (S1) and impressions (S2) (e.g., consistent with imaging features of malignant tumor). This chain must exhibit complete logical derivation and clinical diagnostic value.

**Scoring Criteria.** We employ GPT-4o to score the FC, IC, and LRC sub-chains along multiple dimensions. All metrics are rated on a 10-point scale.

The scoring criteria of FC are shown in Table 12.

The scoring criteria of IC are shown in Table 13.

Table 13: Impression Chain (IC) scoring criteria.

| Chain Level | Scoring Dimension | Description | Scoring Criteria |
|---|---|---|---|
| Impression Chain | Clarity | Medical clarity of the impression statement. | **10**: Impression is completely clear, logically coherent, and unambiguous. |
| | | | **8–9**: Description is clear with only minor incomplete expressions or slight ambiguity (<10%). |
| | | | **6–7**: Basically clear but contains some ambiguities that moderately hinder understanding. |
| | | | **4–5**: Blurry description with numerous ambiguities significantly affecting medical interpretation. |
| | | | **1–3**: Very difficult to understand; almost unusable due to severe ambiguity. |
| | | | **0**: Impression chain is empty or entirely invalid and unclear. |
| | Consistency | Logical consistency of the impression with the underlying FINDING chain. | **10**: Fully consistent with the factual chain; all impressions are derived from facts with no unreasonable content. |
| | | | **8–9**: Overall consistent with only minor (<10%) deviations from the factual chain. |
| | | | **6–7**: Partially inconsistent with the factual chain, showing moderate deviation (10–30%). |
| | | | **4–5**: Large proportion of content inconsistent with or weakly related to the factual chain (30–50%). |
| | | | **1–3**: Impression content is mostly illogical and unrelated to the factual chain. |
| | | | **0**: Impression is entirely invalid or completely contradicts the factual chain. |
| | Medical Utility | Clinical usefulness of the impression for diagnosis and decision-making. | **10**: Impression chain is highly useful, directly supporting diagnosis and clinical decision-making with no additional input needed. |
| | | | **8–9**: High clinical utility; most content is medically meaningful with only minor adjustments required. |
| | | | **6–7**: Partial diagnostic value but considerable portions lack utility or have vague meaning (10–30%). |
| | | | **4–5**: Very limited clinical utility (>50% of content lacks diagnostic value). |
| | | | **1–3**: Impression provides almost no diagnostic significance or is largely incorrect. |
| | | | **0**: Impression chain is invalid or contains no medically meaningful statements. |

The scoring criteria of LRC are shown in Table 14.

**Scoring Procedure.** *Input:* The ground-truth (GT) reasoning chain and the model prediction (Pred) are provided to a scoring large language model (LLM). *Processing:* Guided by a set of predefined scoring rules embedded in the evaluation prompt, the scoring LLM compares the two chains and performs a detailed analysis. *Output:* The LLM produces a structured JSON object containing the classified chains and the numerical scores for each metric, for example:{ "scoring": { "s1_finding": { "existence_match": "8/10", "completeness": "7/10", "accuracy": "9/10" }, "s2_impression": {...}, "s3_reasoning": {...}, "overall_score": "xx/100" } }

**Overall Score Computation.** The final chain-of-thought evaluation score (cot_e) is calculated as a weighted average of the sub-chain scores, balancing the relative importance of different reasoning levels:

$$CoT_e = W_{FC} \cdot \frac{1}{N} \sum_{0}^{N} (S_{FC}^i) + W_{IC} \cdot \frac{1}{N} \sum_{0}^{N} (S_{IC}^i) + W_{LRC} \cdot \frac{1}{N} \sum_{0}^{N} (S_{LRC}^i), \qquad (9)$$

Table 14: Long Reasoning Chain (LRC) scoring criteria.

| Chain Level | Scoring Dimension | Description | Scoring Criteria |
|---|---|---|---|
| Long Reasoning Chain | Logical Completeness | Logical closure and completeness of higher-order reasoning. | **10**: The reasoning chain perfectly covers all key points, with no logical gaps or omissions. |
| | | | **8–9**: Reasoning is largely complete, with only minor (<10%) logical gaps or omitted details. |
| | | | **6–7**: Some logical interruptions or omissions exist, but most reasoning paths remain valid (10–30%). |
| | | | **4–5**: Significant logical gaps, missing many key points, and notable interruptions in reasoning (30–50%). |
| | | | **1–3**: Most of the reasoning chain is invalid; significant logical flaws, key derivations incomplete. |
| | | | **0**: No reasoning process or the reasoning chain completely fails. |
| | Reasoning Depth | Whether the reasoning depth reflects cross-entity and hierarchical associations. | **10**: Reasoning demonstrates highly complex hierarchical relationships and deep cross-entity connections. |
| | | | **8–9**: Reasoning shows moderate depth; most steps are reasonable with minor (<10%) missing complexity. |
| | | | **6–7**: Reasoning depth is insufficient; logical chains are relatively shallow, capturing only surface-level inference (10–30% missing depth). |
| | | | **4–5**: Reasoning lacks depth, limited to single-layer derivations or simple restatements of facts. |
| | | | **1–3**: Reasoning is very superficial; most content invalid or lacks analytical depth. |
| | | | **0**: Reasoning chain has no depth; no higher-order inference. |
| | Clinical Relevance | Whether the reasoning contributes to diagnosis and aligns with medical context. | **10**: Reasoning fully aligns with medical context and is highly relevant and practical. |
| | | | **8–9**: Most reasoning is medically meaningful; only minor content is irrelevant (<10%). |
| | | | **6–7**: Some reasoning entries are meaningful, but overall relevance is limited (10–30% invalid content). |
| | | | **4–5**: Majority of content lacks medical significance; only a few entries provide support (>30% clinically irrelevant). |
| | | | **1–3**: Reasoning is almost clinically useless; content shows deviation from medical background. |
| | | | **0**: Reasoning chain is entirely meaningless or invalid. |
| | Evidence Integration | Whether multiple findings and cues are integrated reasonably. | **10**: Reasoning seamlessly integrates all evidence from finding/impression chains, supporting conclusions. |
| | | | **8–9**: Most evidence is integrated, with minor (<10%) gaps or weak concentration. |
| | | | **6–7**: Partial integration; some information not adopted or weakly related (10–30%). |
| | | | **4–5**: Integration is poor; reasoning is limited to single evidence items (>30% weak integration). |
| | | | **1–3**: Integration is largely insufficient; evidence shows no clear relation. |
| | | | **0**: Reasoning completely detached from evidence; no integration or logical coherence. |

where $S_{FC}^i$ is the score the sample $i$ in FC scoring, $S_{IC}^i$ is the score the sample $i$ in FC scoring, $S_{LRC}^i$ is the score the sample $i$ in FC scoring, and $N$ represents sample number.

The weighting coefficients satisfy:

$$W_{FC} + W_{IC} + W_{LRC} = 1,$$

where $W$ can be used to adjust the importance of reasoning chains during evaluation. In this study, we set $W_{FC} = 0.3$, $W_{IC} = 0.3$, and $W_{LRC} = 0.4$, emphasizing the importance of high-level logical reasoning (LRC) while assigning equal foundational weight to factual findings (FC) and medical impressions (IC).

## G  CASES STUDY

### G.1  TUMORCHAIN HALLUCINATION CORRECTION

In this section, we present a comparative case study to evaluate the clinical utility of TumorChain, focusing on its performance in CoT report generation. We compare the generated reports of our proposed TumorChain with those of a 2D medical model (HealthGPT) and a generalist model (Qwen2.5-VL).

Figure 10 showcase the report outputs of the three models across a clinical scenario of a pancreatic malignant tumor, and explicitly contrast each output with the ground truth. In this example, TumorChain-7B accurately identified the location of the pancreatic tumor, abnormal density within the pancreatic parenchyma, and key features such as the capsule boundary. Although HealthGPT was extensively finetuned on general medical datasets and possesses strong image analysis capabilities, it misdiagnosed the tumor's location, incorrectly identifying it as originating from the posterior instead of the head of the pancreas. Qwen2.5-VL, meanwhile, provided an erroneous diagnosis of no abnormality in the pancreatic head.

This case demonstrates that our dataset construction process—extracting original report information by organ substructure—significantly improves the model's ability to detect abnormalities within organ subregions.

### G.2  QUALITATIVE ERROR TAXONOMY

In table 1 in our paper, Tasks 1–3 correspond to fine-grained VQA pairs targeting lesion attributes. The results demonstrate that our model achieves strong performance on single-step, single-attribute tumor questions. To further analyze the sources of errors in open-ended report generation, we randomly selected 100 reports with incorrect conclusions and invited clinical experts to qualitatively review each failed case. This analysis aims to identify major error patterns and determine which lesion attributes are more prone to mistakes that could affect the final diagnosis. The main error types and representative examples are summarized as follows:

• **False positives due to similar CT appearances:** For example, in pancreatic cases, abnormalities such as pancreatitis or pancreatic pseudocysts are often misclassified as malignant tumors because some malignancies can also present as inflammatory changes.

• **Missed small lesions at organ boundaries due to anatomical overlap:** On 3D CT images, rare lesion morphologies at organ edges or corners are difficult for the model to detect. In one case, a small (approximately 2.0×1.0 cm) mixed-density lesion at the posterior edge of the pancreatic tail was missed because the overall pancreas size and surrounding anatomy appeared normal, causing the model to overlook this subtle abnormality.

• **Incorrect identification of the primary organ due to tumor compression:** In some cases, tumors compress adjacent organs, leading the model to misattribute the origin of the malignancy. For example, a tumor in the pancreatic tail compressing the stomach resulted in the model erroneously diagnosing a malignant lesion in the stomach.

**Question**

Please carefully analyze the specific manifestations of each anatomical part of the pancreas in this CT image. Do you think this patient has pancreatic lesions? Please provide a pancreatic report, including your analytical approach and predicted conclusion.



**HealthGPT**

Based on this CT image, we can observe the specific manifestations of various anatomical parts of the pancreas. In the anterior part of the pancreas, we can see a circular pancreas with a distinct abnormal area, which may be a manifestation of pancreatic lesions. There is also a small abnormal area in the posterior part of the pancreas. These abnormal areas may indicate the presence of pancreatic lesions. Considering the presence of these abnormal areas, we can speculate that the patient has pancreatic lesions. This analysis is based on the obvious manifestations of abnormal areas in the pancreas in CT images, combined with knowledge of pathological diagnosis.

**TumorChain**

**Analysis approach:** Firstly, observe the entire pancreas. A focal solid mass of approximately 17 × 16 mm can be seen in the uncinate process of the pancreatic head, with slightly lower density in the enhanced parenchymal phase, indicating a solid mass with local invasive features. There is no clear damage to the pancreatic capsule or invasion of pancreatic structures, indicating no obvious capsule rupture or extensive infiltration. Secondly, by observing the segmentation of the pancreas, the above-mentioned mass can be seen in the head of the pancreas (uncinate process), and no abnormalities are found in the pancreatic body and tail, indicating that the lesion is limited to the uncinate process of the head of the pancreas. Thirdly, observe the structures around the pancreas. There is no significant contact between the arterial side and the superior mesenteric artery, abdominal trunk, or common hepatic artery. There is no significant contact between the venous side and the portal vein or superior mesenteric vein, but imaging suggests that the contact surface with the superior mesenteric vein is ≤ 180 °; No truncation or dilation of pancreatic duct observed; No suspicious enlargement was found in the surrounding lymph nodes, and no signs of metastasis were observed in the liver and peritoneum. Based on the above findings, it is highly suspected to be a pancreatic malignant tumor, which is consistent with the imaging findings of pancreatic cancer in the head of the pancreas (the focus is about 17 × 16 mm), with local vascular contact (superior mesenteric vein ≤ 180 °), and no distant metastasis.

**Summary:** There is a lesion in the pancreas.

**Qwen2.5-VL**

This CT image shows a cross-sectional view of the abdomen, revealing partial structures of the pancreas. The following is an analysis of various parts of the pancreas:

1. **Pancreatic head:** located in the upper right corner of the image, circular or elliptical in shape, with uniform density and clear boundaries. The head of the pancreas is a part of the pancreas and should not have any obvious abnormalities.
2. **Pancreatic body:** located below the head of the pancreas, in the shape of a long strip, with uniform density and clear boundaries.
3. **Pancreatic tail:** located below the pancreatic body, it is elongated, uniformly dense, and has clear boundaries.

Figure 10: A case of CoT report generation.

