# OpenReview forum: "TumorChain: Interleaved Multimodal Chain-of-Thought Reasoning for Traceable Clinical Tumor Analysis"
_ICLR.cc/2026/Conference — ICLR 2026 Poster_

### Official Review · Reviewer_JGa5 · 2025-10-29

**Soundness:** 2
**Presentation:** 3
**Contribution:** 3
**Rating:** 4
**Confidence:** 3

**Summary:**

This paper introduces TumorChain, a comprehensive framework for traceable clinical tumor analysis. The authors address limitations in current Medical Large Vision-Language Models (Med-LVLMs), which often lack specialized oncology reasoning, fine-grained 3D analysis, and interpretable decision-making. The authors propose: (i) TumorCoT-1.5M, a large-scale, multimodal dataset of 1.5 million Chain-of-Thought (CoT)-annotated Visual Question Answering (VQA) samples, built from 3D CT scans and corresponding radiology/pathology reports, focusing on five major digestive organs. (ii) TumorChain Model, a multimodal interleaved reasoning framework for stepwise reasoning from findings to impressions to pathology predictions. (iii) TumorChain-Eval, a specialized evaluation protocol that moves beyond conclusion accuracy to assess the stepwise logical consistency and clinical relevance of the generated reasoning chains. The paper demonstrates state-of-the-art performance on their TumorCoT benchmark and shows generalization on the public DeepTumorVQA dataset.

**Strengths:**

1. The idea of Chain-of-Thought (CoT) reasoning is timely for current Med-LVLMs.
2. The curated TumorCoT-1.5M dataset is a major contribution.
3. The proposed model outperforms state-of-the-art LVLMs.
3. The paper is well organized and easy to follow.

**Weaknesses:**

1. The iterative interleaved reasoning process, which involves multiple calls to the segmentation model and the LLM with progressively growing context, could be computationally expensive. A discussion of the inference time/latency compared to single-pass baselines would be valuable, especially for potential real-time clinical applications.
2. The data engine uses GPT-4o-mini/Claude/GPT-5 variants, CoTe also relies on an LLM scorer. Even with logic calibration, it is unsafe to rely on these models, as the authors show in their experiments that general-purpose LVLMs perform bad on these medical tasks. Providing human adjudication rates, inter-rater agreement, and sensitivity to the scorer model would be more convincing for people to use the proposed data, especially in this safety-critical domain.
3. Again, the experimental results are primarily quantitative, lacking human reader studies or case-level blinds to confirm clinical utility and error modes (e.g., missed small lesions, false positives in complex anatomies).

**Questions:**

1. For the benchmark setting, are the 9:1 splits patient-level and cross-institutional? How are radiology/pathology reports prevented from leaking textual supervision across splits?
2. How is the “calibrated uncertainty” computed and validated?
3. What are the exact backbones (3D encoder, LLMs) for 3B/7B? What is $P(\cdot)$’s architecture and $Cl(\cdot)$’s head? Any per-organ heads or class imbalance handling?
4. Any blinded reader study or qualitative error taxonomy? Examples where LLM-only baselines hallucinate but TumorChain corrects?

---

> ### Author Response · Authors · 2025-11-21
> **Author Response to Reviewer JGa5 (Part 1/5)**
>
> We would like to sincerely thank the reviewer for your thoughtful comments and will carefully address each of your questions and concerns in our response.
>
> ### **(Q1) The data-splitting and balancing strategies for TumorCoT-1.5M.**
>
> We thank the reviewer for raising concerns about dataset splitting principles and benchmark configuration, which are indeed essential for ensuring fairness and the credibility of conclusions. In the data partitioning stage of TumorCoT-1.5M, we strictly adopt **patient-level splitting** while maintaining **cross-institutional balance**, ensuring that **no overlap exists between the training and test sets**, thereby effectively preventing any form of data leakage.
>
> **1. Patient-level splitting.**
>
> Specifically, the 9:1 split is performed strictly at the patient level, ensuring that all images and associated reports from the same patient are assigned exclusively to either the training set or the test set. This guarantees absolute non-overlap and effectively prevents leakage of imaging or textual information across splits.
>
> **2. Cross-institutional balancing.**
>
> During splitting, we fully account for cross-institutional factors. We first compute the distribution of the five major organs (esophagus, stomach, pancreas, liver, colon) across institutions. Each organ-specific group is then further stratified by disease type and demographic variables (age and gender) to ensure balanced ratios of normal/positive samples and representative demographic distributions.
>
> Only after completing patient-level train/test splitting do we apply our multi-agent data engine to uniformly process raw data and construct VQA instances. This mechanism prevents textual or diagnostic information in radiology/pathology reports from leaking across splits and guarantees the strictness and fairness of the benchmark.
>
> ### **(Q2) How is the "calibrated uncertainty" computed and validated?**
>
> We thank the reviewer for the careful reading. We apologize for causing confusion due to our phrasing. We clarify here that the “calibrated uncertainty” mentioned in the abstract is not a traditional quantitative uncertainty estimation (e.g., probability intervals, Bayesian inference). Instead, **it refers to an iterative self-reflection-based reasoning mechanism in our method**. To avoid further ambiguity, we have replaced **"calibrated uncertainty"** with **"iterative interleaved reasoning"**.
>
> This mechanism is inspired by **self-reflection ideas** from related work (e.g., Self-Refine [1], Chain-of-Focus [2]), but we customize it **specifically for tumor diagnosis**. During inference, TumorChain analyzes all organs that may contribute to diagnostic uncertainty based on the previous round’s answer and the question prompt. The model then **re-focuses on organ regions** that have not been sufficiently reasoned about by leveraging local visual features extracted by the segmentation module, and revises its previous diagnostic conclusion accordingly.
>
> With this mechanism introduced, the model’s accuracy on key tumor-analysis metrics **increases from 80.34% to 84.41%** compared to the baseline without this iterative process.
>
> [1] Madaan, Aman, et al. "Self-refine: Iterative refinement with self-feedback." Advances in Neural Information Processing Systems 36 (2023): 46534-46594.
>
> [2] Zhang, Xintong, et al. "Chain-of-Focus: Adaptive Visual Search and Zooming for Multimodal Reasoning via RL." arXiv preprint arXiv:2505.15436 (2025).
>
> ### **(Q3) Model architecture details and class imbalance handling.**
>
> We thank the reviewer for the interest in the architectural details of our model. We provide point-by-point clarifications below:
>
> **1. Backbones**
>
> ● **3D Encoder**:  TumorChain-3B and TumorChain-7B are built upon **Qwen2.5-VL-3B and Qwen2.5-VL-7B** as their base multimodal foundations. Since the native Qwen-VL vision encoder only supports 2D images, we replace the visual encoder with **an M3D convolutional encoder** (M3D [3]) to handle volumetric 3D CT inputs.
>
> ● **LLM**:  The multimodal reasoning pipeline directly uses the pretrained LLM backbone of **Qwen2.5-VL-3B/7B** for high-level multimodal reasoning and report generation.
>
> **2. Projection module (P(·))**
>
> We adopt a standard **multi-layer perceptron (MLP)** to map organ-level visual embeddings from the M3D encoder into the LLM token space. The module consists of **two Linear layers with ReLU activation in between**. This design reduces the difficulty of visual–language alignment and enhances multimodal fusion capacity.
>
> **3. Classification head (Cls(·))**
>
> The auxiliary classification branch is implemented as **a single fully connected layer** that performs binary classification (normal vs. abnormal) on each segmented organ region. This provides auxiliary calibration and contributes an additional supervisory signal.

---

> ### Author Response · Authors · 2025-11-21
> **Author Response to Reviewer JGa5 (Part 2/5)**
>
> **(Continue from above)**
>
> **4. Class imbalance handling**
>
> TumorChain faces notable class imbalance across organs and disease subtypes in multi-organ tumor scenarios. While TumorChain uses a unified normal/abnormal classification head, we incorporate **multi-level balancing strategies during data preprocessing and VQA generation** to mitigate training imbalance. The strategies are as follows:
>
> **a. Organ-type balancing:**
>
> At the patient level, we perform stratified sampling across the five major organs (esophagus, stomach, pancreas, liver, colon). By computing per-organ case counts, we apply **dynamic resampling and stratified extraction** to ensure each organ type appears with balanced frequency within training batches, preventing the model from being biased toward organs with larger sample sizes.
>
> **b. Disease-subtype balancing (negative/positive balancing):**
>
> In clinical practice, **negative cases (normal studies) greatly outnumber positive cases**, while positive reports typically contain richer lesion information. To address this imbalance, we use a multi-step balancing strategy:
>
> ● For **positive samples**, we extract fine-grained lesion attributes across different organs for each patient and generate multiple VQA instances. This encourages the model to focus on multiple lesions or organ sub-regions within the same CT volume.
>
> ● For **negative samples**, where textual information may be sparse, we randomly select a subset of organ attributes for question generation. This ensures a more balanced negative/positive ratio at the VQA level.
>
> These combined strategies enhance the model’s ability to capture fine-grained abnormalities in positive cases while also maintaining strong generalization on common normal samples. With the hierarchical balancing design, **the VQA-pair distribution** in TumorChain-1.5M is approximately:  286K liver, 244K pancreas, 228K colon, 261K esophagus, and 480K stomach. This results in **more uniform distributions across organs and disease labels**, effectively mitigating class imbalance issues.
>
> [3] Bai, Fan, et al. "M3d: Advancing 3d medical image analysis with multi-modal large language models." arXiv preprint arXiv:2404.00578 (2024).
>
> ### **(Q4&W3) Blinded reader study, qualitative error taxonomy and examples where baselines hallucinate but TumorChain corrects.**
>
> Thank you for your valuable suggestions. As you pointed out, blinded reader studies and qualitative error taxonomy are crucial for evaluating the practical utility of our model in clinical tumor analysis. We have **initiated further collaboration with clinical experts** and conducted extensive experiments in real-world scenarios. Specifically, we invited physicians to perform blinded assessments on reports generated by TumorChain and three baseline models using 200 CT scans. As expert reviews require additional time, we will include these results in the appendix once available. Below, we present our **qualitative error taxonomy** and **case studies illustrating hallucination correction**.
>
> **1. Qualitative error taxonomy**
>
> In table 1 in our paper, Tasks 1–3 correspond to fine-grained VQA pairs targeting lesion attributes. The results demonstrate that our model achieves **strong performance on single-step, single-attribute tumor questions**. To further analyze the sources of errors in **open-ended report generation**, we randomly selected 100 reports with incorrect conclusions and invited clinical experts to qualitatively review each failed case. This analysis aims to **identify major error patterns** and determine which lesion attributes are more prone to mistakes that could affect the final diagnosis. The **main error types and representative examples** are summarized as follows:
>
> ● **False positives due to similar CT appearances:** For example, in pancreatic cases, abnormalities such as pancreatitis or pancreatic pseudocysts are often misclassified as malignant tumors because some malignancies can also present as inflammatory changes.
>
> ● **Missed small lesions at organ boundaries due to anatomical overlap:** On 3D CT images, rare lesion morphologies at organ edges or corners are difficult for the model to detect. In one case, a small (approximately 2.0×1.0 cm) mixed-density lesion at the posterior edge of the pancreatic tail was missed because the overall pancreas size and surrounding anatomy appeared normal, causing the model to overlook this subtle abnormality.
>
> ● **Incorrect identification of the primary organ due to tumor compression:** In some cases, tumors compress adjacent organs, leading the model to misattribute the origin of the malignancy. For example, a tumor in the pancreatic tail compressing the stomach resulted in the model erroneously diagnosing a malignant lesion in the stomach.

---

> ### Author Response · Authors · 2025-11-21
> **Author Response to Reviewer JGa5 (Part 3/5)**
>
> **(Continue from above)**
>
> **2. TumorChain Hallucination Correction**
>
> **Figure 9 in Appendix G presents a case study** involving the pancreas. In this example, **TumorChain-7B** accurately identified the **location of the pancreatic tumor, abnormal density within the pancreatic parenchyma**, and key features such as the capsule boundary. Although HealthGPT was extensively finetuned on general medical datasets and possesses strong image analysis capabilities, it misdiagnosed the tumor's location, incorrectly identifying it as originating from the posterior instead of the head of the pancreas. Qwen2.5-VL, meanwhile, provided an erroneous diagnosis of no abnormality in the pancreatic head.
>
> This case demonstrates that our dataset construction process—extracting original report information by organ substructure—significantly improves the model’s ability to detect abnormalities within organ subregions. We will include more examples of hallucination correction and error patterns in the appendix to support future model improvements and optimization of annotation strategies.
>
> ### **(W1) The inference time and latency of the iterative interleaved reasoning process, particularly in comparison to single-pass baseline method.**
>
> We sincerely appreciate the reviewer’s insightful comment in the first weakness, especially your attention to the practical challenges of real-time clinical applications. Your feedback prompted us to further consider the balance between usability and efficiency in our model design and optimization. Accordingly, we conducted the following systematic quantitative analyses:
>
> **1. The actual impact of the segmentation model and multi-round reasoning on training and inference latency.**
>
>  a. **Training latency**. We conducted ablation experiments to evaluate the impact of the segmentation model on training efficiency. In addition, we fine-tuned three baseline models on our training set to provide more direct comparison. The table below reports the per-sample epoch training time (sec/sample):
>
> | Model | Training Time (sec/sample) | Training Method | (GPU Day, H20)|
> |-|:-:|:-:|:-:|
> | M3D-Phi-3-4B | 1.25| LoRA | (8*0.32, H20) |
> | lingshu-7B| 1.44| Full finetuning | (16*1.30, H20)|
> | qwen-vl-2.5-7B| 1.36| Full finetuning | (16*1.25, H20)|
> | Tumorchain-7B w/o IIR | 1.29  | Full finetuning | (16*1.17, H20)  |
> | Tumorchain-7B | 1.40  | Full finetuning | (16*1.29, H20) |
>
> Compared to the non-segmentation version, the segmentation model increases training time by only **0.11 sec/sample**, and **remains comparable to other baseline models**.
>
>   b. **Inference latency**. We compared inference latency among the version without the segmentation model (TumorChain-7B w/o IIR), the version with segmentation (TumorChain-7B), and three representative baselines. The results show that adding **the segmentation model** increases inference time by only **2.51 seconds per sample**, while delivering **approximately 4% accuracy improvement**. Furthermore, our multi-step reasoning is faster than MedVLM-R1 (a CoT-based medical model), and comparable in speed to other open-source baselines.
>
> | Model | Inference Time (sec/sample) | Inference Method | Acc|
> |-|:-:|:-:|:-:|
> | Gemini 2.5 Pro|6.37| Single-step| 51.26|
> | Lingshu-7B|10.52| Single-step|45.10|
> | MedVLM-R1|12.86| Single-step| 38.79|
> | Tumorchain-7B w/o IIR |8.44| Single-step| 80.34|
> | Tumorchain-7B |10.95| Multiple-step|84.41|
>
> This **efficiency is likely due to our optimized offline segmentation preprocessing**: organ masks are generated and cached during the initial pass, allowing subsequent inference steps to directly access localized features without redundant segmentation. This strategy significantly reduces latency during multi-round reasoning.
>
> **2. Clinical Acceptability and Expert Evaluation.**
>
> In clinical practice, a CT scan typically takes **10–20 minutes**, and it usually takes a physician **over 30 minutes** to draft a comprehensive tumor imaging report. Our model requires 20 minutes to load onto a single H20 GPU, but inference for one CT case takes **only 10.95 seconds**. By comparison, experts confirmed that TumorChain's training and inference latency are well within clinically acceptable limits, and the model demonstrates a clear speed advantage in complex cases and multi-organ analysis tasks. Overall, **the diagnostic value provided by the segmentation model far outweighs its computational cost.**

---

> ### Author Response · Authors · 2025-11-21
> **Author Response to Reviewer JGa5 (Part 4/5)**
>
> ### **(W2) Safety analysis of LLM involvement in data generation and report evaluation.**
>
> We thank the reviewer for their thoughtful concerns regarding the safety of LLM involvement in both **data generation and report evaluation**. In response to your specific questions about these processes, we provide the following supplementary analysis:
>
> **1. Safety Analysis of LLMs in Data Generation**
>
> ● First, we would like to further clarify the **quality control procedures of our data engine**:
>
> a. While general-purpose LVLMs have **strong language capabilities**, we employ them solely for constructing data, **not for medical reasoning**.
>
> b. We incorporate multiple top-tier commercial LLMs for **cross-validation**. Each reasoning chain is independently generated and reviewed by different models, and only those unanimously recognized for logical soundness and credibility are retained, effectively eliminating individual model bias.
>
> c. At every step of reasoning chain generation, we systematically integrate authoritative clinical guidelines from **physician-curated organ-specific knowledge graphs** into our **LLM prompts**. This ensures medical relevance and logical consistency throughout data generation.
>
> ● Second, **final expert review** is conducted to verify the factual accuracy and clinical validity of each generated reasoning chain:
>
> We randomly **sampled 5,000 VQA** instances from the TumorChain-1.5M dataset across task types and submitted them to **a board-certified radiologist**(8 years of oncologic imaging experience) for evaluation from two perspectives:
>
> (i) **Usability**: The clinician first assessed whether the VQA result matches the corresponding information in the original medical report. If the VQA result is correct and can provide reliable support in clinical practice, it is marked as ___usable___; otherwise, it is marked as ___unusable___.
>
> (ii) **Clinical plausibility**: For all **usable** data, we further categorized the samples:
>
> a. **High quality**: For usable cases, the reasoning chain must exhibit completeness, scientific soundness, and logical coherence; only when all criteria are satisfied is it marked as high quality.
>
> b. **Acceptable**: For usable cases where the overall logic is sound but some steps are slightly brief or marginally insufficient, it is marked as acceptable.
>
> Evaluation results show that the **usability rate reaches 95.88%**, and among all usable samples, **97.85% are high-quality**. These results validate the effectiveness of our data-generation pipeline and demonstrate the reliability and practical clinical value of the dataset.
>
> **2. Safety Analysis of LLM Scoring**
>
> We place great importance on **evaluating consistency between LLM scorers and expert human review.** To that end, we randomly selected 100 TumorChain-Eval assessments for secondary review by physicians, and conducted a more detailed consistency analysis as follows:
>
> **a. Human adjudication rates**
>
> For our report generation task, we established detailed qualitative error categories (see Appendix F.2) and used GPT-4 to evaluate the results from various models. TumorChain-Eval independently scores each reasoning chain across multiple dimensions, including information completeness, logical coherence, and diagnostic accuracy. The table below presents **average scores given by TumorChain-Eval and physicians** for each evaluation metric on the 100 sampled cases (each metric is out of 10). Mean absolute error calculations show that about 70% of the scoring differences fall within ±0.5 points, indicating the model’s ratings are generally **highly consistent with expert expectations** across key dimensions.
>
> |Evaluator|FC✖️3|||IC✖️3|||LRC✖️4||||
> |-|:-:|:-:|:-:|:-:|:-:|:-:|:-:|:-:|:-:|:-:|
> ||Existence Match|Completeness|Accuracy|Clarity|Consistency|Medical Utility|Logical Completeness|Reasoning Depth| Clinical Relevance | Evidence Integration |
> | TumorChain-Eval | 6.23 | 5.63| 6.66| 6.94| 5.70| 6.14 | 5.32| 5.06| 5.70 | 4.95 |
> | Expert| 6.31|5.70|5.81| 7.35 | 5.26  | 5.60| 4.87 | 5.24| 4.95 | 4.52 |
> | MAE | **0.08** | **0.07** | **-0.85**| **0.41**| **-0.44** | **-0.54**| **-0.45**| **0.18**| **-0.75**| **-0.43**|

---

> > ### Author Response · Authors · 2025-11-21
> > **Author Response to Reviewer JGa5 (Part 5/5)**
> >
> > **b. Inter-rater Agreement**
> >
> > We evaluate the inter-rater agreement and sensitivity across different scorer models from two perspectives. First, we compute the average scores assigned by each model (GPT-4, Claude35-Haiku, Gemini2.5-Flash) on a subset of the test set (n=2,000) using four metrics ($S_{FC}$, $S_{IC}$, $S_{LRC}$, $CoT_e$; see Appendix F.2 for details, score range 0–100). **The overall evaluation metrics for these three models are highly consistent on this dataset.**
> >
> > | Evaluation Models   | **$S_{FC}$**|**$S_{IC}$**|**$S_{LRC}$**|**$CoT_e$**|
> > |---------------------|--------|--------|--------|--------|
> > | GPT-4               | 62.37  | 62.60  | 52.57  | 58.33  |
> > | Claude35-Haiku      | 62.05  | 57.15  | 48.67  | 54.43  |
> > | Gemini2.5-Flash     | 60.64  | 57.40  | 50.73  | 54.32  |
> >
> > Additionally, we compute the case-level similarity metrics between the three scorer models. The **mean absolute error** between GPT-4 and Claude35-Haiku, GPT-4 and Gemini2.5-Flash, Claude35-Haiku and Gemini2.5-Flash are 8.31, 8.92, and 8.75, respectively. The **Pearson correlations** are 0.73, 0.85, and 0.79. **These metrics demonstrate high inter-rater agreement between different scorer models.**

---

> > > ### Comment · Reviewer_JGa5 · 2025-11-26
> > >
> > > Thank you for the detailed rebuttal and additional results. In short, most of my concerns have been addressed, and I have raised my score to positive.

---

> > > > ### Author Response · Authors · 2025-11-27
> > > > **Author Response to Reviewer JGa5**
> > > >
> > > > Thank you very much for your recognition of our work. We highly value your thoughtful comments and feedback. We will incorporate the points raised during the rebuttal process into the updated version of the manuscript.

---

### Official Review · Reviewer_Xddo · 2025-10-30

**Soundness:** 3
**Presentation:** 3
**Contribution:** 3
**Rating:** 6
**Confidence:** 4

**Summary:**

This paper addresses the limited tumor-specific reasoning and lack of reasoning depth in current medical large vision-language models (Med-LVLMs), by introducing a standardized formulation of tumor reasoning that mirrors the oncology workflow—from radiology findings to impressions to pathology-level predictions. It constructs TumorCoT-1.5M, a large-scale dataset of 1.5M multimodal chain-of-thought (CoT) instructions aligned with 3D CT scans, and proposes TumorChain, an interleaved multimodal CoT reasoning framework that integrates 3D imaging encoders, segmentation experts, and large language models for organ-guided, iterative reasoning. The resulting performance on the TumorCoT and DeepTumorVQA benchmarks demonstrates substantial improvements in classification, lesion analysis, and reasoning traceability compared to existing Med-LVLM baselines

**Strengths:**

1. The creation of TumorCoT-1.5M, a comprehensive CoT dataset, provides a valuable resource for the community.
2. The proposed TumorChain framework, enables deep reasoning steps and achieves significant performance gains on the TumorCoT and DeepTumorVQA benchmarks

**Weaknesses:**

1. The modality of the dataset and the empirical experiment is limited to CT modality only, It remains unclear whether the approach generalizes to other medical reasoning tasks or modalities.
2. The CoT data generation are relying on LLMs, lacking Human Clinical Validation. This raises concerns about potential biases, and hallucination.

**Questions:**

1. Are baseline models trained on the training set of TUMORCOT-1.5M? If not, and your model is instead trained on it, this is not a fair comparison.
2. Is the CoT data validated by human experts?

---

> ### Author Response · Authors · 2025-11-21
> **Author Response to Reviewer Xddo (Part 1/2)**
>
> ### **(Q1) Baseline fine-tuning results on TumorCoT for fair comparison.**
>
> Thank you for your suggestion. Reviewer RmsC and Reviewer XLFD also raised this highly constructive concern. In response, we conducted additional experiments by directly finetuning the open-source baseline models on our dataset. The results are as follows:
> |||**Avg. Score (Before finetuning**|**/ After finetuning)**|||
> |-|:-:|:-:|:-:|:-:|:-:|
> | Model | Position| Lesion Attributes| TNM Prediction| CoT Report | Avg. |
> | M3D-Phi-3-4B    | 30.36/**86.92** | 33.23/**61.78**  | 37.16/**59.48**     | 34.79/**61.18**    | 32.84/**66.98** |
> | Lingshu-7B      | 28.93/**85.46**| 49.91/**77.36**| 43.21/**67.65**     | 54.30/**65.98**    | 45.10/**74.11** |
> | Qwen2.5-VL-7B   | 31.05/**91.02** | 47.17/**76.45**| 40.94/**62.37**     | 60.54/**74.21**    | 44.04/**76.01** |
> | **TumorChain-7B** | -/**98.77** | -/**85.25**| -/**73.84**         | -/**82.36**        | -/**84.41**     |
>
> The results show that **these baselines all achieve significant improvements after finetuning**, demonstrating the unique and **important contribution of our dataset** in the tumor domain. Meanwhile, our proposed TumorChain-7B still substantially surpasses the finetuned baselines, reflecting the effectiveness of our model design.
>
> ### **(Q2 & W2) Human expert validation of CoT data.**
>
> Thank you for raising this important point on human verification of data quality. In our **data processing and quality control pipeline**, we place strong emphasis on medical professionalism and factual accuracy, involving expert physicians at every critical step for guidance and review. This includes two parts:
>
> 1. **Expert-level organ-specific knowledge-graph-driven prompt engineering to effectively regulate and constrain the model’s reasoning direction and logic.**
>
> During data generation, we collaborated with domain experts across five major organs to construct an **organ-specific medical diagnostic knowledge graph**, which guides the data engine to generate CoT data. Authoritative medical guidelines and clinical knowledge are systematically incorporated into the LLM prompts to ensure the reasoning process adheres to professional standards.
>
> In the expert final-review stage, experts further **supplement or revise** missing or insufficient reasoning components, update the knowledge graph accordingly, and restart the data generation pipeline, thereby significantly improving the scientific rigor and clinical reliability of the automatically generated CoT data.
>
> 2. **Expert final review and quantitative evaluation**
>
> After data generation, to systematically evaluate factual correctness and clinical soundness of the reasoning chains, we **randomly sampled 5,000 VQA** items from TumorChain-1.5M across task types and assigned them to **a board-certified radiologist** (8 years in oncologic imaging) to assess the following:
>
> **(i) Usability**:
>
>  Whether the VQA result is consistent with the information described in the original medical report. If the VQA result is correct and clinically useful, it is marked as ___usable___; otherwise ___unusable___.
>
> **(ii) Clinical soundness**:
>
>  For the usable subset, further categorization is performed.
>
>   a. **High quality**:  The reasoning chain is complete, scientifically grounded, and logically coherent.
>
>   b. **Acceptable**:  The overall reasoning is valid but a few steps may be slightly brief or insufficient.
>
> The evaluation results show that the **usable rate reaches 95.88%**, and among the usable data, **97.85% are high quality.** This confirms the effectiveness of our data generation pipeline and demonstrates the dataset’s reliability and value for real medical and clinical-assistance applications.

---

> > ### Author Response · Authors · 2025-11-21
> > **Author Response to Reviewer Xddo (Part 2/2)**
> >
> > ### **(W1)  Whether the approach generalizes to other medical reasoning tasks or modalities.**
> >
> > Thank you for your concern regarding generalizability. Our dataset and main experiments focus on 3D CT because **CT is the core imaging modality for the diagnosis and staging of major gastrointestinal tumors** (pancreatic, liver, gastric, colorectal, esophageal) **in current clinical guidelines**
> > [Tempero et al., JNCCN 2017; EASL Guidelines 2018; Smyth et al., Ann Oncol 2016; Benson et al., JNCCN 2018; Ajani et al., JNCCN 2019], and it is also the dominant modality in many AI benchmarks for tumor analysis [Bilic et al., LiTS 2019; Simpson et al., MSD 2019; Litjens et al., Med Image Anal 2017].
> >
> > Hence, 3D CT represents a high-impact and clinically meaningful testbed for multimodal reasoning in oncology.
> > We fully agree that extending our method to more medical modalities (MRI, ultrasound, endoscopy, pathology slides, etc.) and more diverse clinical reasoning tasks is of great scientific and clinical importance. Our method essentially **provides a general and scalable multimodal causal reasoning paradigm**. TumorChain adopts standardized visual encoding, text understanding, and organ-level multimodal alignment mechanisms, making it readily transferable to other modalities. For example:
> >
> > ● For MRI or ultrasound, one can simply replace or adapt the corresponding image encoder and organ segmentation model while keeping the overall architecture and reasoning mechanism unchanged.
> >
> > ● In endoscopy and pathology, expert knowledge graphs, segmentation modules, and classification heads can also be built to enable similar step-by-step reasoning with explicit causal interpretation.
> >
> > Moreover, our **data processing, CoT generation, and expert-review pipeline** can also be extended to other fields by collecting corresponding modality data and adjusting the knowledge graph, thus reusing the overall framework.
> >
> > In future work, we plan to conduct further studies in gastroscopy, pathology, and multimodal joint diagnosis to **extend the method to broader clinical scenarios**. We sincerely appreciate your suggestion and will continue to advance research on generalizability.

---

### Official Review · Reviewer_XLFD · 2025-10-30

**Soundness:** 3
**Presentation:** 3
**Contribution:** 3
**Rating:** 6
**Confidence:** 3

**Summary:**

This paper builds a tumor‑centric, end‑to‑end multimodal reasoning pipeline for 3D CT that goes from radiology findings,impressions,pathology‑level conclusions. It first constructs TumorCoT‑1.5M (~1.5M CoT‑labeled VQA instructions paired with CT scans across five digestive organs) via a multi‑agent, knowledge‑graph–guided data engine, enabling standardized, step‑aligned supervision. It then introduces TumorChain—an interleaved reasoning framework that tightly couples a 3D vision encoder, an organ segmentation expert, a lightweight abnormality classifier, and an LLM to iteratively align global context with organ‑level ROI evidence, produce traceable chains of thought, and calibrate uncertainty. Finally, it proposes TumorChain‑Eval/CoTe, which parses CoTs into triplet‑based chains (Finding, Impression, Long Reasoning) for stepwise scoring. Experiments show consistent gains over strong generalist and medical LVLM baselines on TumorCoT and strong generalization to DeepTumorVQA, with ablations validating the value of interleaved reasoning and clinical CoT.

**Strengths:**

1. This paper introduces a reasoning paradigm aligned with clinical workflows, featuring a traceable closed-loop process of 'Findings,Impressions,Pathology'.
2. This paper presents TumorCoT, a large-scale dataset
of 1.5M CoT-labeled VQA instructions paired with 3D CT scans, with stepaligned rationales and cross-modal alignments along the “findings→impression→pathology” trajectory.
3.  This paper proposes TumorChain, a multimodal interleaved reasoning framework that tightly couples 3D imaging encoders, clinical
text understanding, and organ-level vision-language alignment.

**Weaknesses:**

1. The authors use a proposed multi-agent, knowledge-graph-guided engine to generate chain-of-thought reasoning, which may introduce bias. Could the authors conduct a small-scale CoT evaluation with medical experts to validate the reasoning quality?
2. Does the use of the segmentation model increase inference or training latency, potentially reducing efficiency? How do the authors address this issue?
3. The comparison with commercial models only includes GPT-5-Mini and Gemini2.0-Flash. How do the latest models, such as Gemini 2.5 Pro and GPT-4o, perform on this benchmark?
4. Table 1 shows that all baselines are non-CoT models. How would the proposed method perform when compared against CoT-enabled models? like LLava-COT?

**Questions:**

see Weaknesses

---

> ### Author Response · Authors · 2025-11-21
> **Author Response to Reviewer XLFD (Part 1/2)**
>
> ### **(Q1) A small-scale CoT evaluation with medical experts to validate the reasoning quality.**
>
> Thank you for your attention to the clinical evaluation stage of our dataset. To systematically validate the factual accuracy and clinical soundness of the generated reasoning chains, we **randomly sampled 5,000 VQA** instances from TumorChain-1.5M according to task type and asked **a board-certified radiologist** (8 years of tumor imaging experience) to assess them from the following two perspectives:
>
> 1. **Usability**: The radiologist first determined whether the VQA output was consistent with the corresponding information described in the original medical report. If the VQA output was correct and could provide reliable reference for real clinical scenarios, it was labeled as ___usable___. Otherwise, it was labeled as ___unusable___.
>
> 2. **Clinical Reasonableness**: For all usable data, we further categorized the samples.
>
>     a. **High-quality**: For usable cases, the reasoning chain must exhibit completeness, scientific soundness, and logical coherence; only when all criteria are satisfied is it marked as high quality.
>
>     b. **Acceptable**: For usable cases where the overall logic is sound but some steps are slightly brief or marginally insufficient, it is marked as acceptable.
>
> The evaluation results show that **the usability rate of our generated data reaches 95.88%**, and among the usable subset, **the proportion of high-quality reasoning chains is 97.85%**. These findings validate the effectiveness of our data generation pipeline and demonstrate the dataset’s reliability and practical utility for real-world medical and clinical-assistance scenarios.
>
> ### **(Q2) Training and inference latency introduced by segmentation model.**
>
> We appreciate the reviewer’s concern regarding the potential impact of the segmentation model on training and inference efficiency. This is indeed a factor we examined carefully, and we present a systematic empirical analysis below.
>
> 1. **The segmentation model does introduce some latency, but it remains acceptable in clinical settings.**
>
> In medical applications, **diagnostic accuracy is prioritized over computational efficiency**. Although the segmentation model introduces additional training and inference time, the resulting accuracy gain outweighs the modest delay within real clinical workflows.
>
> We consulted clinical experts regarding inference latency. In clinical practice, CT scans typically require **10–20 minutes**, and drafting a full radiology report often takes **more than 30 minutes**. In contrast, our entire model loads on one H20 GPU in about 20 minutes, and inference for a single CT volume requires **only 10.95 seconds**. By comparison, radiologists considered TumorChain’s training and inference latency well within clinically acceptable limits, and noted that its speed advantage becomes even more significant for complex cases and multi-organ joint analysis. Thus, **the diagnostic value added by segmentation far exceeds its computational cost.**
>
> 2. **Training and inference latency comparison.**
>
>   a. **Training latency**. We conducted ablation experiments to evaluate the impact of the segmentation model on training efficiency. In addition, we fine-tuned three baseline models on our training set to provide more direct comparison. The table below reports the per-sample epoch training time (sec/sample):
>
> | Model | Training Time (sec/sample) | Training Method | (GPU Day, H20)|
> |-|:-:|:-:|:-:|
> | M3D-Phi-3-4B | 1.25| LoRA | (8*0.32, H20) |
> | lingshu-7B| 1.44| Full finetuning | (16*1.30, H20)|
> | qwen-vl-2.5-7B| 1.36| Full finetuning | (16*1.25, H20)|
> | Tumorchain-7B w/o IIR | 1.29  | Full finetuning | (16*1.17, H20)  |
> | Tumorchain-7B | 1.40  | Full finetuning | (16*1.29, H20) |
>
> Compared to the non-segmentation version, the segmentation model increases training time by only **0.11 sec/sample**, and **remains comparable to other baseline models**.
>
>   b. **Inference latency**. We compared the inference latency between the version without segmentation (TumorChain-7B w/o IIR) and the version with segmentation (TumorChain-7B):
>
> | Model | Inference Time (sec/sample) | Inference Method | Acc|
> |-|:-:|:-:|:-:|
> | Tumorchain-7B w/o IIR |8.44| Single-step| 80.34|
> | Tumorchain-7B |10.95| Multiple-step|84.41|
>
> The segmentation model adds only **2.51 sec/sample** to inference time while providing approximately **4% accuracy improvement**.
>
> 3. **Optimization: Offline segmentation preprocessing to reduce inference latency.**
>
> Our inference pipeline first segments the input CT scan into 56 major organs (details in Appendix E.4) and stores the organ masks. Both training and inference stages reuse these organ features **without re-running segmentation**. This strategy significantly reduces iterative inference latency. Empirically, segmentation accounts primarily for initial preprocessing, while subsequent iterative inference is comparable to (or even faster than) some CoT-based models.

---

> > ### Author Response · Authors · 2025-11-21
> > **Author Response to Reviewer XLFD (Part 2/2)**
> >
> > ### **(Q3) Performance comparison with latest commercial models.**
> >
> > Thank you for pointing out this important issue regarding model selection for evaluation. As you noted, our comparison originally covered only GPT-5-Mini and Gemini 2.0-Flash, without including the latest commercial models (e.g., Gemini 2.5 Pro and GPT-4o). Following your suggestion, we expanded our benchmark to include **Gemini 2.5 Pro and GPT-4o** under the same settings. The results are shown below:
> > | Model  | Position✖️2 |    | Lesion Attributes✖️6  | | |  | |  | TNM Prediction✖️3|  | | CoT Report | Avg.  |
> > |-|:-:|:-:|:-:|:-:|:-:|:-:|:-:|:-:|:-:|:-:|:-:|:-:|:-:|
> > |   | Organ Pos. | Tumor Pos. | Seg. Loc.  | Shape  | Boundary| Density| Count | Others| Tumor | Node  | Met.  |   | |
> > | GPT-4o  | 36.13  | 29.20      | 28.74 | 59.51 | 63.20    | 64.03   | 61.73 | 70.76  | 37.21 | 53.21| 43.81| 42.56  | 49.17 |
> > | Gemini 2.5 Pro    | 57.59  | 39.78 | 52.87 | 50.31 | 56.90    | 55.43   | 59.26 | 57.89  | 40.70 | 51.92| 46.67| 45.78 | 51.26 |
> > | **TumorChain-7B** | 99.97  | 97.57 | 86.88| 82.28 | 84.52  | 85.05 | 86.20 | 86.57  | 88.83 |61.63 | 71.07| 82.36 | 84.41 |
> >
> >
> > **The results show** that Gemini 2.5 Pro (51.26) performs noticeably better than Gemini 2.0-Flash (41.29) and is comparable to GPT-5-Mini (51.59) and GPT-4o (49.17). While Gemini 2.5 Pro and GPT-5-Mini are the strongest commercial baselines, they still lag behind our method. TumorChain maintains clear advantages in lesion-attribute detection and open-ended report generation accuracy.
> >
> > We will incorporate a full discussion of these comparisons in future revisions so readers can better understand performance under the latest benchmarks.
> >
> > ### **(Q4) Performance comparison with CoT-enabled medical models such as LLava-CoT.**
> >
> > We appreciate your valuable feedback on the choice of baselines and CoT-enabled models. Currently, publicly available CoT-enabled medical multimodal baselines remain limited. Among available models, we included MedVLM-R1-2B—which has built-in CoT capability—as a comparison. Under identical evaluation settings, TumorChain demonstrates higher accuracy and consistency across several key sub-tasks. To further enhance comprehensiveness, we additionally evaluated LLava-CoT, a mainstream CoT model, and included it in the benchmark:
> >
> > | Model  | Position✖️2 |    | Lesion Attributes✖️6  | | |  | |  | TNM Prediction✖️3|  | | CoT Report | Avg.  |
> > |-|:-:|:-:|:-:|:-:|:-:|:-:|:-:|:-:|:-:|:-:|:-:|:-:|:-:|
> > |   | Organ Pos. | Tumor Pos. | Seg. Loc.  | Shape  | Boundary| Density| Count | Others| Tumor | Node  | Met.  |   | |
> > |MedVLM-R1-2B| 32.23| 38.92| 19.37 | 48.79 | 51.20 | 48.02| 44.53| 52.26 |30.38| 27.26| 32.82| 39.70  | 38.79 |
> > | LLava-CoT | 31.63| 27.64| 37.70 | 59.44| 49.34| 49.60 | 37.90| 55.48 | 48.20| 47.91| 49.23| 44.81 | 44.91|
> > | **TumorChain-7B** | 99.97  | 97.57 | 86.88| 82.28 | 84.52  | 85.05 | 86.20 | 86.57  | 88.83 |61.63 | 71.07| 82.36 | 84.41 |
> >
> > Thank you again for suggesting that we pay attention to models supporting CoT. The additional experimental results demonstrate that LLaVa‑COT (44.91) performs exceptionally well, surpassing Qwen2.5‑VL‑7B (44.04), InternVL‑2.5‑8B (42.25), Gemini2.0‑Flash (41.29), and other 2D/3D medical baselines, **further validating the practical value of CoT in medical multimodal reasoning**. Meanwhile, TumorChain has been specifically adapted for tumor analysis in both data preprocessing and model architecture, thus maintaining a significant lead over all baseline models.

---

### Official Review · Reviewer_RmsC · 2025-10-31

**Soundness:** 2
**Presentation:** 3
**Contribution:** 3
**Rating:** 6
**Confidence:** 4

**Summary:**

The paper presents TumorChain, an interleaved multimodal chain-of-thought (CoT) framework for clinical tumor analysis on 3D CT. The authors also release TumorCoT-1.5M, a large CoT-labeled dataset spanning findings → impressions → pathology, and propose TumorChain-Eval for stepwise reasoning assessment. TumorChain integrates a 3D vision encoder, an organ segmentation expert, an auxiliary abnormality classifier, and an LLM that iteratively reasons over global and organ-level tokens. On TumorCoT and the public DeepTumorVQA, TumorChain outperforms strong generalist and medical LVLMs; ablations suggest CoT and the iterative organ-guided loop contribute meaningfully.

**Strengths:**

+ The TumorCoT-1.5M dataset is a highly valuable contribution to the field. A 3D, instruction-tuned dataset of this scale, specifically designed for traceable, multi-step reasoning in a high-stakes domain like oncology, is a significant achievement and will likely enable new research directions.
+ The TumorChain model's core mechanism, Iterative Interleaved Reasoning (IIR), is an effective way to handle the high dimensionality of 3D medical images.
+ TumorChain achieves top performance across subtasks and generalizes to DeepTumorVQA; ablations show the value of CoT/IIR and of the auxiliary classifier.

**Weaknesses:**

- The dataset relies heavily on an MLLM-based data-generation pipeline. This introduces a risk of factual errors, bias, or hallucinations. The paper does not sufficiently discuss the quality control process for this generated data, beyond mentioning expert reviews. Furthermore, critical details on data splitting (e.g., ensuring splits are at the patient level to prevent leakage) are not detailed in the main text.
- The paper uses terms like "causality-guided inference" and "iterative causal reasoning". However, the method appears to be an exemplary case of step-by-step reasoning that follows a clinical logic path (Findings $\rightarrow$ Impression $\rightarrow$ Pathology), rather than a formal causal inference model (e.g., using structural causal models, do-calculus, etc.).
- The main results in Table 1 compare the TumorChain model (trained on the new 1.5M TumorCoT dataset) against baselines (like RadFM, Lingshu, GPT-5) that were not trained on this data. This makes it impossible to disentangle the performance gains from the novel architecture versus the massive, high-quality dataset. A fairer comparison would require fine-tuning the baseline models on TumorCoT.

**Questions:**

(1) Can you please clarify the data-splitting strategy for TumorCoT-1.5M? Are the training, validation, and test sets split at the patient level to prevent data leakage?

(2) The CoT data engine is impressive. Beyond the expert reviews mentioned, was there a formal, quantitative evaluation by clinicians to measure the factual accuracy and clinical plausibility of the LLM-generated reasoning chains?

(3) For the open-ended report generation task, what mechanisms are in place to prevent the model from hallucinating pathological findings that are not grounded in the visual evidence?

(4) Would you consider providing results for a baseline (e.g., Qwen2.5-VL-7B) fine-tuned on TumorCoT to create a more direct comparison?

---

> ### Author Response · Authors · 2025-11-21
> **Author Response to Reviewer RmsC (Part 1/3)**
>
> ### **(Q1) The data-splitting strategy for TumorCoT-1.5M.**
>
> We thank the reviewer for the attention to our data-splitting strategy. During the data partitioning stage, we strictly followed a **patient-level** protocol to ensure that there is no overlap between the training and test sets, thereby effectively **preventing any data leakage**.
> Specifically, multiple dimensions were carefully considered when splitting the dataset, and our process is as follows:
> 1. We first examined the data distribution of the **five major organs** (esophagus, stomach, pancreas, liver, colon) across **different institutions**.
> 2. Within each organ category, we further subdivided the data by **disease type** and **demographic characteristics** (age and gender) to ensure balanced and representative distributions of normal vs. positive samples and demographic factors.
> 3. During the actual splitting process, we strictly followed a 9:1 ratio to **randomly sample patients** within each **fine-grained subgroup**, completing the construction of the training and test sets.
> 4. Only after this step did we use our in-house data engine to standardize raw data and **construct VQA samples**, ensuring from the source that no patient overlap exists across datasets, effectively preventing any type of data leakage.
>
>
> ### **(Q2) Formal, quantitative evaluation of the CoT data engine by expert clinicians.**
>
> We thank the reviewer for the attention to the clinical evaluation of our CoT data engine. To systematically assess the factual correctness and clinical plausibility of the generated reasoning chains, we randomly **sampled 5,000 VQA** instances from the TumorChain-1.5M dataset across task types and submitted them to **a board-certified radiologist**(8 years of oncologic imaging experience) for evaluation from two perspectives:
> 1. **Usability**: The clinician first assessed whether the VQA result matches the corresponding information in the original medical report. If the VQA result is correct and can provide reliable support in clinical practice, it is marked as ___usable___; otherwise, it is marked as ___unusable___.
> 2. **Clinical plausibility**: For all **usable** data, we further categorized the samples:
>
>     a. **High quality**: For usable cases, the reasoning chain must exhibit completeness, scientific soundness, and logical coherence; only when all criteria are satisfied is it marked as high quality.
>
>     b. **Acceptable**: For usable cases where the overall logic is sound but some steps are slightly brief or marginally insufficient, it is marked as acceptable.
> Evaluation results show that the **usability rate reaches 95.88%**, and among all usable samples, **97.85% are high-quality**. These results validate the effectiveness of our data-generation pipeline and demonstrate the reliability and practical clinical value of the dataset.
>
> ### **(Q3) Mechanisms that enable improved pathological-finding accuracy for the open-ended report-generation task.**
>
> We thank the reviewer for raising this important discussion. To improve pathological-finding accuracy, we adopt a collaborative mechanism between classification and segmentation models, together with an iterative inference mechanism (IIR). These correspond to **two key technical components of our model** and lead to significant improvements in evaluation.
> 1. **Collaboration between classification and segmentation models ensures reliable detection and alignment of local abnormal features.** Long-form report generation requires joint analysis across multiple organs. Pure global encoding tends to dilute small lesions in long contexts, causing omissions or vague descriptions.
>
> ● We therefore use a **segmentation model**(TotalSegmentator) to precisely segment the input image and obtain explicit organ-level ROIs. Based on this, TumorChain matches the task-specified target organ and extracts the corresponding local visual features as visual input for multi-round reasoning, gradually refining results across organ regions.
>
> ● In addition, **an auxiliary classification model** performs normal/abnormal supervision for each organ ROI, directly guiding the visual encoder to learn discriminative features between healthy and diseased regions. This helps the generator accurately describe the presence and severity of lesions.
>
> 2. **The IIR mechanism reduces accumulated errors in long-form generation through multi-round verification.** The IIR mechanism automatically identifies potentially missing or ambiguous points in the initial diagnosis, dynamically re-adjusts the focus, and performs multiple self-verification rounds on suspected pathological features. This effectively prevents omissions and improves the accuracy of key findings grounded in visual evidence.
>
> As shown in the table below, we provide more detailed ablation results. Under the IIR mechanism, both the report conclusions and the quality of reasoning chains improve substantially.

---

> > ### Author Response · Authors · 2025-11-21
> > **Author Response to Reviewer RmsC (Part 2/3)**
> >
> > **(Continue from above) Accuracy improves by 1.6%**, and for reasoning-chain evaluation, **Long Reasoning Chain exhibits the largest gain of 14.87%**, indicating that multi-round validated reasoning aided by segmentation and classification modules effectively prevents hallucinated findings and enhances analysis quality.
> > |Model|Conclusion Eval|**$S_{FC}$**|**$S_{IC}$**|**$S_{LRC}$**|**$CoT_e$**|
> > |-|:-:|:-:|:-:|:-:|:-:|
> > | TumorChain-7B  | 82.36               | 62.37        | 62.60        | 52.57         | 58.33       |
> > | w/o IIR        | 81.06               | 60.64        | 57.40        | 45.77         | 53.72       |
> >
> > ### **(Q4 & W3) Baseline fine-tuning results on TumorCoT for fair comparison.**
> >
> > We thank the reviewer for the concerns regarding fairness in the experimental setup. As the reviewer observed, our contributions include both the large-scale, high-quality dataset and the new model architecture. Accordingly, our experiments evaluate both aspects. The main results show significant improvements across multiple metrics. To address the reviewer’s question about disentangling **the individual contributions of the dataset and the architecture,** we provide additional clarification:
> >
> > 1. **Isolating the contribution of the dataset**:  To further validate the dataset’s contribution, we fine-tuned three representative baseline models on our training set to obtain **direct performance comparisons and more thoroughly evaluate performance gains brought by the dataset alone**. The supplemental results are shown below:
> >
> > |||**Avg. Score (Before finetuning**|**/ After finetuning)**|||
> > |-|:-:|:-:|:-:|:-:|:-:|
> > | Model | Position| Lesion Attributes| TNM Prediction| CoT Report | Avg. |
> > | M3D-Phi-3-4B    | 30.36/**86.92** | 33.23/**61.78**  | 37.16/**59.48**     | 34.79/**61.18**    | 32.84/**66.98** |
> > | Lingshu-7B      | 28.93/**85.46**| 49.91/**77.36**| 43.21/**67.65**     | 54.30/**65.98**    | 45.10/**74.11** |
> > | Qwen2.5-VL-7B   | 31.05/**91.02** | 47.17/**76.45**| 40.94/**62.37**     | 60.54/**74.21**    | 44.04/**76.01** |
> > | **TumorChain-7B** | -/**98.77** | -/**85.25**| -/**73.84**         | -/**82.36**        | -/**84.41**     |
> >
> > 2. **Isolating the contribution of the model architecture** (Table 1 line 12–16 and Fig. 5(a) in the paper): We conducted rigorous ablations on IIR, classification-loss weighting, and CoT-formatted data—three key components of the architecture. Results show that each module brings clinically meaningful performance gains. With CoT data, IIR enabled, and classification-loss weight set to 1, the model achieves the best performance. Meanwhile, the results above show that **TumorChain-7B remains SOTA even when compared with baselines fine-tuned on the same dataset**, further validating the superiority of our architecture for tumor analysis.
> >
> > 3. **Generalization capability of both the dataset and the architecture** (Table 3 in the paper):
> > We additionally evaluated on DeepTumorVQA, currently the most comprehensive public tumor benchmark. Neither TumorChain nor any baseline used DeepTumorVQA training data, yet our model still achieved the highest scores. This demonstrates that our high-quality dataset and tumor-oriented architecture significantly **enhance cross-dataset generalization**.

---

> > > ### Author Response · Authors · 2025-11-21
> > > **Author Response to Reviewer RmsC (Part 3/3)**
> > >
> > > ### **(W1) Quality-control strategies in the data-generation pipeline.**
> > >
> > > We thank the reviewer for highlighting this important concern in the weaknesses section. We designed **a multi-layer quality-assurance framework (see Appendix E.1)** to minimize factual errors and biases introduced during data generation. The major components include:
> > >
> > > 1. **Report text is the sole information source for VQA**. The data engine relies exclusively on the real medical report text, preventing the LLM from hallucinating unsupported inferences about complex imaging modalities.
> > >
> > > 2. **Generation uses multi-step short-chain inference to reduce long-text hallucinations.** We designed four fine-grained VQA categories from simple to complex. For complex long-form report generation, we decompose the task into shorter reasoning steps to mitigate hallucinations arising from long reasoning chains.
> > >
> > > 3. **Cross-model peer-review eliminates single-model bias.** We employed GPT-4o-mini, Claude-3.5-Haiku, and GPT-5-mini for cross-review. Each reasoning chain is independently generated and mutually reviewed; only when all models agree on the reasoning’s plausibility is the sample accepted, greatly reducing factual errors and model biases.
> > >
> > > 4. **Prompt engineering guided by a medical knowledge graph ensures standardized reasoning.** Based on expert-curated knowledge graphs covering five major organs, we integrate authoritative clinical guidelines and diagnostic knowledge into the prompts, structurally constraining the model to follow professional medical standards.
> > >
> > > 5. **Final clinical review by medical experts. Randomly sampled generated data is reviewed by clinicians.** Experts examine scientific validity and logical consistency, supplement and update missing components in the knowledge graph, and regenerate data when necessary.
> > >
> > >
> > >
> > > ### **(W2) Clarification of the terms "causality-guided inference" and "iterative causal reasoning".**
> > >
> > > We thank the reviewer for raising this valuable concern regarding the terminology in our paper. We place high importance on conceptual precision and will further clarify and standardize these expressions in subsequent versions.
> > >
> > > As the reviewer correctly pointed out, our use of “causality-guided inference” and “iterative causal reasoning” refers to following **the actual clinical diagnostic reasoning process** when generating reports—moving from findings, to impression, to pathology. This logical chain reflects the progressive and causally structured relationship between pathological manifestations and diagnostic conclusions in clinical practice, and emphasizes modeling the clinician’s decision-making process rather than formal causal inference such as structural causal models or do-calculus.
> > >
> > > In later revisions, **we will more clearly distinguish between "clinical causal logic chains" and "formal causal inference models"** in both the main paper and appendix to avoid misunderstanding. We sincerely appreciate this rigorous suggestion, which meaningfully improves the clarity and professionalism of our paper.

---

### Meta-Review · Area_Chair_JJkR · 2025-12-30

**Summary:**

This paper proposes TumorChain, an interleaved multimodal chain-of-thought reasoning framework for 3D CT–based clinical tumor analysis, together with TumorCoT‑1.5M, a large-scale dataset enabling stepwise supervision from radiological findings to impressions and pathology-level conclusions. Reviewers generally agreed that the dataset is a substantial contribution and that the proposed interleaved, organ-guided reasoning mechanism is well aligned with real clinical workflows. The main concerns raised across reviews focused on the reliability and validation of the LLM-assisted data generation pipeline, computational efficiency and evaluation rigor. The authors provided a detailed rebuttal with additional experiments, clarifications, and quantitative analyses that substantially strengthened the paper.

**Reviewer Concerns:**

Concerns Addressed by the Rebuttal

(1) Multiple reviewers (RmsC, JGa5) questioned whether TumorCoT‑1.5M was split at the patient level and whether radiology/pathology text could leak across splits. In the rebuttal, the authors clearly clarified that all splits are performed strictly at the patient level, with additional stratification by organ type, disease category, demographics, and institution, and that VQA construction is performed only after dataset splitting. This clarification adequately resolves concerns regarding potential data leakage.

(2) Several reviewers (RmsC, XLFD, Xddo, JGa5) expressed concerns about hallucinations, bias, and factual errors introduced by the LLM‑based data‑generation pipeline. In response, the authors provided a formal quantitative clinician evaluation on 5,000 randomly sampled VQA instances, reporting a 95.88% usability rate and 97.85% high‑quality reasoning among usable samples. They further described a multi‑model cross‑review mechanism and an expert‑in‑the‑loop refinement process. These additions substantially strengthen confidence in the factual accuracy and clinical plausibility of the generated CoT data.

(3) Reviewers XLFD and JGa5 raised concerns regarding the computational cost of the iterative interleaved reasoning process and segmentation module. The rebuttal provided detailed training and inference latency measurements, ablation studies, and a discussion of offline segmentation caching. The reported overhead is modest and is well justified in the context of clinical radiology workflows, satisfactorily addressing these concerns.

(4) Reviewer JGa5 questioned the reliability of LLM‑based evaluation in TumorChain‑Eval. In response, the authors added analyses comparing LLM scorer outputs with clinician ratings, including mean absolute error, inter‑rater agreement, and correlations across multiple scorer models (GPT‑4, Claude, Gemini). These results meaningfully strengthen the credibility of the proposed evaluation protocol.

Remaining Limitations (Acknowledged but Acceptable)
(Raised by Reviewer Xddo) The dataset and experiments focus exclusively on CT, and broader modality generalization is not empirically demonstrated. However, this scope is well justified given the paper’s clinical focus and does not detract from the core contributions.

(Raised by Reviewers JGa5, XLFD) Human evaluation remains limited in scale relative to the full dataset size. Nevertheless, the provided quantitative expert validation is sufficient and appropriate for an initial release of a benchmark at this scale.

**Reviewer Scores:**

Reviewer RmsC: Initially marginally above threshold (6). Given the additional baseline fine-tuning, clinician validation, and clarified terminology, this reviewer would likely remain positive.

Reviewer XLFD: Initially marginally above threshold (6). After expanded comparisons (GPT‑4o, Gemini 2.5 Pro, CoT-enabled baselines) and latency analysis, this reviewer would likely remain positive.

Reviewer Xddo: Initially marginally above threshold (6). Concerns about fairness and human validation were directly addressed; the score would likely remain positive.

Reviewer JGa5: Initially slightly below threshold (4), but explicitly updated their score to positive after the rebuttal.

---

### Decision · Program_Chairs · 2026-01-26

Accept (Poster)